# Intracellular cholesterol trafficking is dependent upon NPC2 interaction with lysobisphosphatidic acid

Leslie A McCauliff[1,2†], Annette Langan[1,2†], Ran Li[1,2†], Olga Ilnytska[1,2], Debosreeta Bose[1,2], Miriam Waghalter[1], Kimberly Lai[1], Peter C Kahn[3], Judith Storch[1,2]*

[1]Department of Nutritional Sciences, Rutgers University, New Brunswick, United States; [2]Rutgers Center for Lipid Research, Rutgers University, New Brunswick, United States; [3]Department of Biochemistry and Microbiology, Rutgers University, New Brunswick, United States

*For correspondence:
storch@sebs.rutgers.edu

†These authors contributed equally to this work

Competing interests: The authors declare that no competing interests exist.

**Abstract** Unesterified cholesterol accumulation in the late endosomal/lysosomal (LE/LY) compartment is the cellular hallmark of Niemann-Pick C (NPC) disease, caused by defects in the genes encoding NPC1 or NPC2. We previously reported the dramatic stimulation of NPC2 cholesterol transport rates to and from model membranes by the LE/LY phospholipid lysobisphosphatidic acid (LBPA). It had been previously shown that enrichment of NPC1-deficient cells with LBPA results in cholesterol clearance. Here we demonstrate that LBPA enrichment in human NPC2-deficient cells, either directly or via its biosynthetic precursor phosphtidylglycerol (PG), is entirely ineffective, indicating an obligate functional interaction between NPC2 and LBPA in cholesterol trafficking. We further demonstrate that NPC2 interacts directly with LBPA and identify the NPC2 hydrophobic knob domain as the site of interaction. Together these studies reveal a heretofore unknown step of intracellular cholesterol trafficking which is critically dependent upon the interaction of LBPA with functional NPC2 protein.

DOI: https://doi.org/10.7554/eLife.50832.001

## Introduction

Cholesterol is a small, hydrophobic molecule that is a vital building block of cell membranes and a precursor for steroid hormones, bile salts, vitamin D, and oxysterol ligands for transcription factors. Intracellular transport of cholesterol is a highly regulated but, as yet, incompletely understood process. Perturbations can lead to detrimental outcomes such as in the lysosomal storage disorder Niemann Pick Type C (NPC) disease, where LDL-derived cholesterol becomes trapped within the late endosomal/lysosomal (LE/LY) system. The sterol particularly enriches LE/LY inner membranes which develop during endosome maturation as a means of compartmentalizing its contents (*Gruenberg, 2001*; *Gruenberg, 2003*; *Matsuo et al., 2004*). The accumulation of cholesterol in NPC disease is associated with amassing of other lipids in the LE/LY, disruption of post-lysosomal cholesterol metabolism, and ultimately clinical manifestations including organomegaly and neurological deterioration. In 95% of NPC cases, mutations in the large LE/LY transmembrane protein, NPC1, prevent proper export of cholesterol from the LE/LY to other cellular compartments. The remaining 5% of cases are caused by mutations in the small, 132-amino acid, soluble LE/LY protein, NPC2 (*Peake and Vance, 2010*; *Vanier, 2010*).

Similarities in the cellular and clinical phenotypes resulting from either NPC1 or NPC2 deficiency led to the suggestion that these two proteins function cooperatively in normal LE/LY cholesterol trafficking (*Kwon et al., 2009*; *Sleat et al., 2004*); a proposed model shows cholesterol directly

**eLife digest** Cholesterol is a type of fat that is essential for many processes in the body, such as repairing damaged cells and producing certain hormones. Normally, cholesterol enters cells from the bloodstream and is then moved to the parts of the cell that need it via a process known as 'trafficking'.

When cholesterol trafficking goes wrong, abnormally large amounts of cholesterol and other fats accumulate within the cell. Over time, these fatty deposits become toxic to cells and eventually damage the affected tissues. Niemann-Pick type C disease (NPC) is a severe genetic disorder affecting cholesterol trafficking. It is characterized by cholesterol build-up in multiple tissues, including the brain, which ultimately causes degeneration and death of nerve cells.

Two proteins, NPC1 and NPC2, are involved in NPC disease. Both proteins normally help move cholesterol out of important trafficking compartments (known as the endosomal and lysosomal compartments) to other areas of the cell where it is needed. Patients with the disease can have mutations in either the gene for NPC1 or the gene for NPC2. This means that cells from NPC1 patients do not make enough functional NPC1 protein (but contain working NPC2), and vice versa.

Previous studies had shown that giving cells with NPC1 mutations large amounts of the small molecule lysobisphosphatidic acid (LBPA for short) could compensate for the loss of NPC1, and stop the toxic build-up of cholesterol. McCauliff, Langan, Li et al. therefore wanted to explore exactly how LBPA was doing this. They had shown that LBPA dramatically increased the ability of purified NPC2 protein to transport cholesterol, and wondered if the effect of LBPA in the cells without NPC1 depended on NPC2. They predicted that boosting LBPA levels would not work in cells lacking NPC2.

Biochemical experiments using purified protein showed that LBPA and NPC2 did indeed interact directly with each other. Systematically changing different building blocks of NPC2 revealed that a single region of the protein is sensitive to LBPA, and when this region was altered, LBPA could no longer interact with NPC2. Since LBPA is naturally produced by cells, they then stimulated cells grown in the laboratory to generate more LBPA using its precursor phosphatidylglycerol. They used cells from patients with mutations in either NPC1 or NPC2 and demonstrated that LBPA's ability to reverse the accumulation of cholesterol was dependent on its interaction with NPC2. Thus, increasing LBPA levels in cells from patients with NPC1 mutations was beneficial, but had no effect on cells from patients with NPC2 mutations.

These results shed new light not only on how cells transport cholesterol, but also on potential methods to combat disorders of cellular cholesterol trafficking. In the future, LBPA could be developed as a genetically tailored, patient-specific therapy for diseases like NPC.

DOI: https://doi.org/10.7554/eLife.50832.002

transferred from NPC2, in the LE/LY lumen, to NPC1, located in the limiting membrane of the compartment (*Estiu et al., 2013*; *Wang et al., 2010*). Recent tertiary structural analyses identifying a potential NPC2 interacting domain on the NPC1 protein support this mode of cholesterol egress from the LE/LY compartment (*Zhao et al., 2016*). It is further proposed that cholesterol transfer from NPC2 to the luminally localized N-terminal domain of NPC1 allows sterol passage through the glycocalyx found at the luminal surface of the LE/LY limiting membrane, via concerted effort from membrane glycoproteins (*Li et al., 2016*).

NPC2 binds cholesterol with a 1:1 stoichiometry (*Xu et al., 2007*). *Ko et al. (2003)* showed that cholesterol binding was necessary but not sufficient for its cholesterol efflux function, and we later demonstrated that cholesterol transport was the second critical component of NPC2 functionality (*Cheruku et al., 2006*; *Xu et al., 2008*). We further showed that cholesterol transfer by NPC2 occurs via protein-membrane interactions (*Cheruku et al., 2006*; *Xu et al., 2008*). Using model membranes, we demonstrated a remarkable, order of magnitude stimulation in wild type NPC2 cholesterol transfer rates by the incorporation of lysobisphosphatidic acid (LBPA), also known as bis-monoacylglycerol phosphate (*Cheruku et al., 2006*; *Xu et al., 2008*). LBPA is a structural isomer of phosphatidylglyerol with an atypical phospholipid stereoconfiguration. It is localized primarily to inner LE/LY membranes and is thought to be involved not only in the formation of these internal

membranes and their architecture, but also in the sorting and efflux of LE/LY components, including cholesterol (*Gruenberg, 2003*; *Hullin-Matsuda et al., 2007*; *Kobayashi et al., 1998*; *Kobayashi et al., 1999*; *Möbius et al., 2003*). Incubation of BHK cells and macrophages with an anti-LBPA monoclonal antibody results in cholesterol accumulation in the LE/LY compartment, resembling the NPC phenotype (*Delton-Vandenbroucke et al., 2007*; *Kobayashi et al., 1999*).

Interestingly, *Chevallier et al. (2008)* showed that by enriching NPC1-deficient cells with exogenously added LBPA, the cholesterol accumulation was reversed ; the underlying molecular mechanism for LBPA stimulated LE/LY cholesterol egress remains unknown. Based on the dramatic effects of LBPA on NPC2-mediated cholesterol transfer, we hypothesized that the mechanism of LBPA action involves its specific functional interaction with NPC2, such that LBPA enrichment of cells deficient in NPC2 would *not* reverse cholesterol accumulation, in contrast to what had been found in NPC1-deficient cells.

In the present studies we demonstrate that, indeed, LBPA enrichment does not lead to the clearance of cholesterol in NPC2-deficient cells despite the presence of functional NPC1. We show for the first time that the mechanism involves an obligate direct interaction of NPC2 with LBPA, identify the LBPA-sensitive domain on the NPC2 surface, and establish the essential functional nature of NPC2-LBPA interactions in cholesterol egress from the LE/LY compartment. The results identify a novel, heretofore unknown step in LE/LY cholesterol egress which is dependent upon LBPA interaction with the hydrophobic knob of NPC2 protein. This in turn suggests that LBPA enrichment may be used to effect cholesterol egress in cells with defective NPC1 but functional WT NPC2, and NPC2 with disease-causing mutations outside the hydrophobic knob domain.

## Results

### Predicted orientation of NPC2 in membranes

Our previous kinetics analyses strongly suggested that the mechanism of cholesterol transfer between NPC2 and membranes was via protein-membrane interaction (*Cheruku et al., 2006*; *McCauliff et al., 2015*; *Xu et al., 2008*). NPC2 does not contain any apparent transmembrane domains, nor are there experimentally documented membrane interactive domains to date. For de novo predictions we therefore employed the Orientation of Proteins in Membranes (OPM) Database, a curated online resource that predicts the spatial positions of known protein structures relative to the hydrophobic core of a lipid bilayer (*Lomize et al., 2012*). Using the crystal structure of bovine NPC2 (PDB ID: 1NEP), a loop domain consisting of hydrophobic residues as highlighted in *Figure 1* was predicted to be highly membrane interactive, with a ΔG of −4.6 kcal/mol. This domain corresponds to 56-HGIVMGIPV-64 and consists primarily of the hydrophobic residues I58, V59, M60, I62, P63, and V64, plus the non-polar residues G61 and G57. Structurally, this domain forms a hydrophobic knob which presents prominently on the surface of the NPC2 protein. OPM predicts that this knob domain inserts into the hydrophobic space of the membrane model, positioning the sterol binding pocket of NPC2 near the membrane surface (*Figure 1A*) and, thus, in proximity to membrane sterols. These computational observations suggest that this knob domain may play a key role in the mechanism by which NPC2 is able to transport cholesterol between inner membranes of the LE/LY compartment. Of note, and as shown in *Figure 1B*, the primary sequence of the hydrophobic knob domain is conserved in mammalian NPC2 proteins but not in the yeast NPC2 homologue. Hydrophobicity scales for NPC2 as well as a Kyte-Doolittle plot of the hydrophobicity scores along the primary protein sequence indicate that the hydrophobicity of the knob domain is conserved amongst mammalian NPC2 proteins, in contrast to the low hydrophobicity in the yeast NPC2 protein (*Figure 1C*).

### LBPA markedly stimulates sterol transfer rates between NPC2 and membranes

NPC2 is a soluble protein but we demonstrated using tryptophan quenching that it is membrane-interactive and that cholesterol transfer between NPC2 and membranes occurs via transient protein-membrane interactions (*Xu et al., 2008*). We previously found that incorporation of 25 mol% LBPA in egg phosphatidylcholine (EPC) membranes resulted in cholesterol transfer rates from vesicles to NPC2 that were markedly accelerated relative to 100% EPC membranes (*Cheruku et al., 2006*;

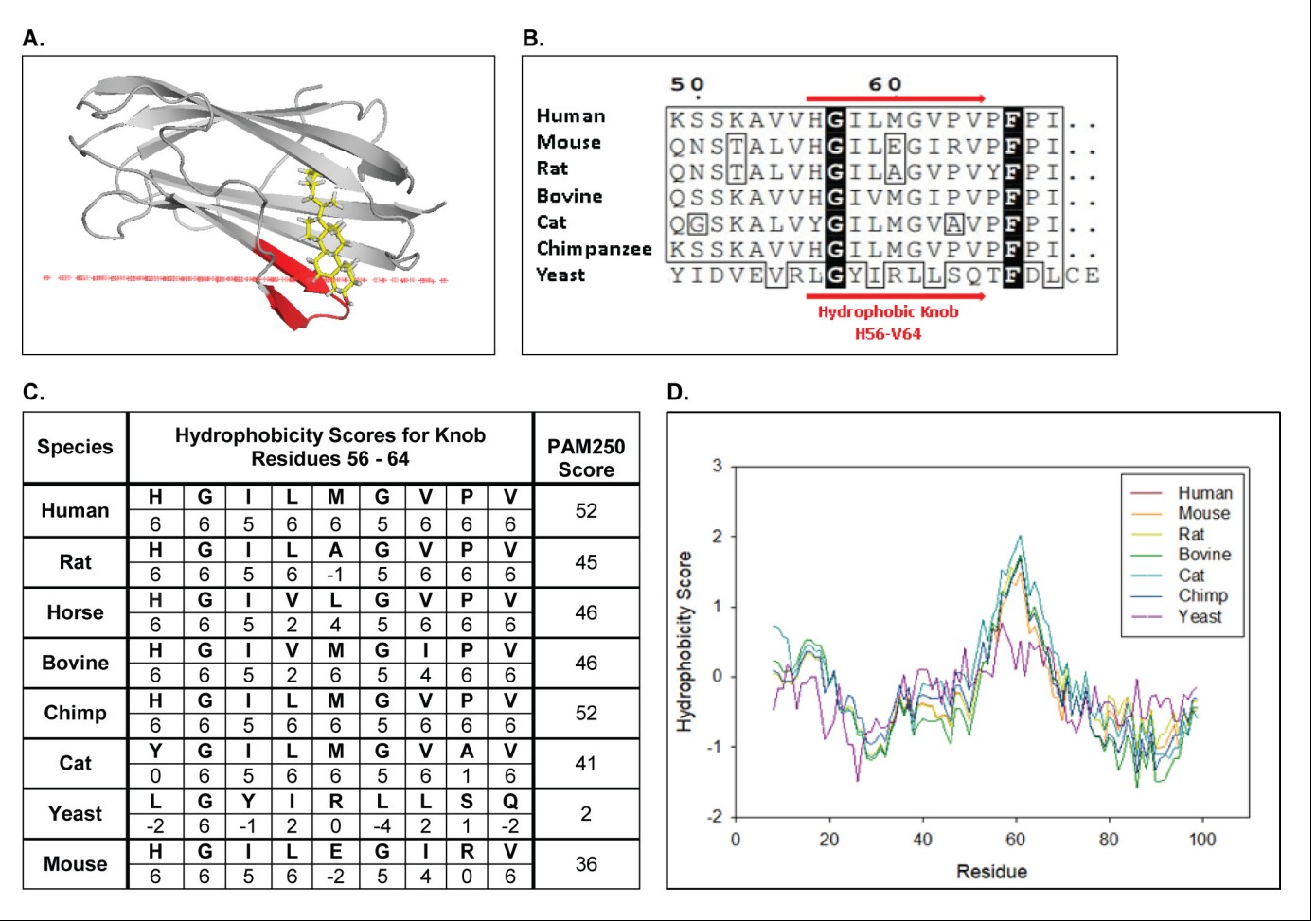

**Figure 1.** OPM predicts NPC2 stably inserts into membranes via a conserved hydrophobic knob region. (A) Holo bovine NPC2 (PDB ID: 2HKA) is predicted by OPM to insert its prominent hydrophobic knob region (red ribbon) into the hydrophobic space of a model membrane, positioning the cholesterol in the sterol binding pocket in close proximity to the membrane surface (red line). (B) Multiple sequence alignment of human NPC2 (NCBI Accession: NP_006423.1), rat NPC2 (NP_775141.2 ) mouse NPC2 (NCBI Accession: NP_075898.1), bovine NPC2 (NCBI Accession: NP_776343.1), cat NPC2 (XP_003987882.1), chimpanzee NPC2 (NP_001009075.1) and the yeast NPC2 (NCBI Accession: KZV12184.1) were aligned with CLUSTAL Omega, and alignment for hydrophobic knob residues H56 to V64 are shown. Consensus sequences are in black and conserved residues are boxed. (C) Conservation scores and hydrophobicity scores for the hydrophobic knob, residues H56 to V64, were calculated based on the PAM250 scoring matrix and Kyte and Doolitle Hydrophobicity scale (**Kyte and Doolittle, 1982**). (D) The aligned NPC2 protein sequences were analyzed with the ProtScale Tool on ExPASy server based on the Kyte and Doolittle Amino acid Hydropathicity scale with a frame window of 15 residues (**Gasteiger et al., 2005**).
DOI: https://doi.org/10.7554/eLife.50832.003

*Xu et al., 2008*) LBPA accounts for approximately 15 mol% of total LE/LY phospholipids, with the likelihood of higher lateral concentrations in the highly heterogeneous inner LE/LY membranes (*Kobayashi et al., 2002*). Thus, we examined the rates of cholesterol transfer from NPC2 to membranes as a function of increasing levels of LBPA. The results in *Figure 2* indicate an exponential relationship between the LBPA content of the vesicles and the NPC2 cholesterol transfer rate; increasing the mol% of LBPA in SUV from 0% to 30% effectively increases the NPC2 cholesterol transfer rates by approximately 100 fold.

The transfer of cholesterol between NPC2 and vesicles was also examined using [3]H-cholesterol, where an NPC2-[3]H-cholesterol complex was added to vesicles followed by centrifugal separation. The reverse reaction was also examined, where membranes with [3]H-cholesterol were added to apo NPC2 followed by centrifugation to separate protein and membranes. The results showed that the labeled cholesterol transfers from NPC2 to membranes and from membranes to NPC2, and in agreement with previous experiments using tryptophan quenching to determine sterol distribution,

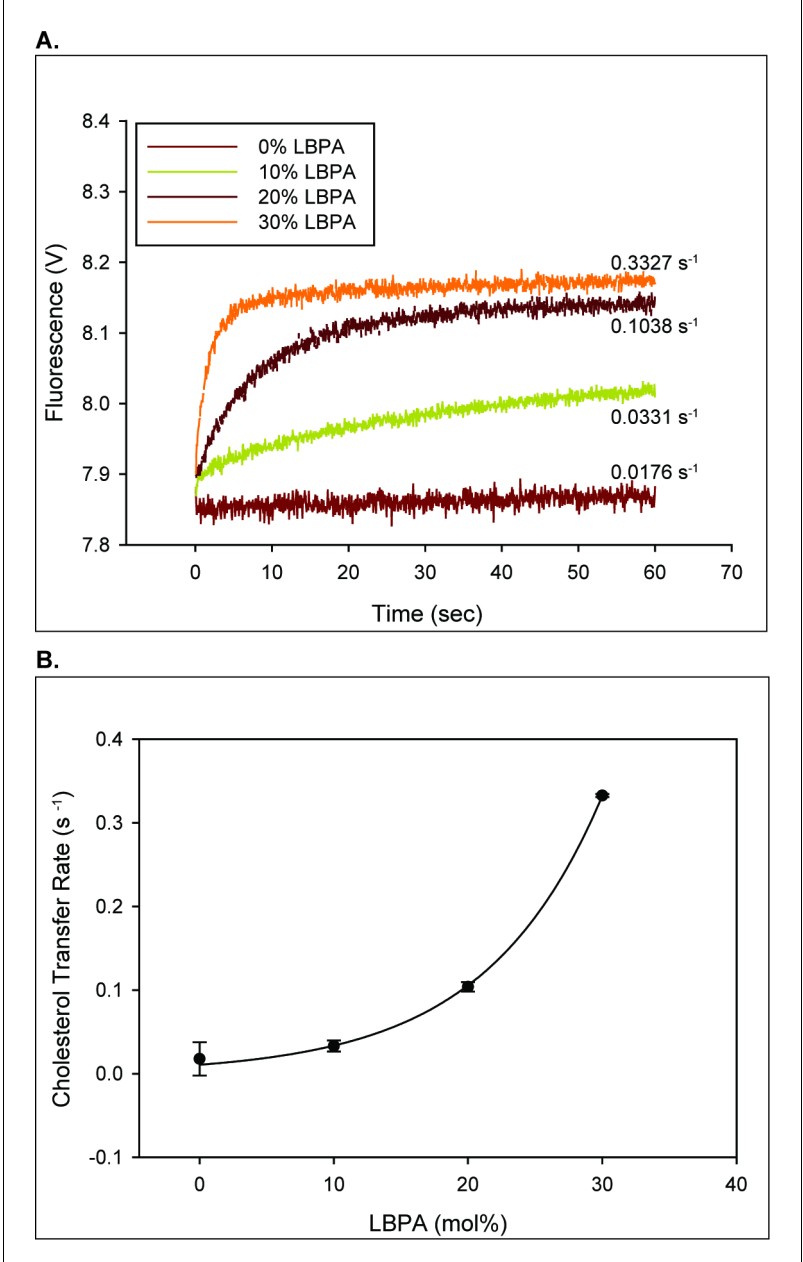

**Figure 2.** LBPA dramatically increase the rate of NPC2 mediated cholesterol transport. (**A**) 2.5 µM NPC2-cholesterol complex was mixed with 250 µM of small unilamellar vesicles containing increasing mole percentages of LBPA in an SX20 stopped flow spectrofluorimeter as described under Materials and methods. The dequenching of endogenous tryptophan fluorescence was used to monitor cholesterol transfer from NPC2 to membranes. (**B**) The transfer rates from NPC2 to small unilamellar vesicles with various mole percentages of LBPA were fitted to a reverse single exponential function in Sigmaplot with an R-squared value of 0.9944. Data are representative of three experiments, each consisting of 2–3 individual runs ± SE.

DOI: https://doi.org/10.7554/eLife.50832.004

the relative partition of cholesterol between NPC2 and phospholipid vesicles was ≥30:1, mol NPC2: mol PL (*Xu et al., 2008*). Since cholesterol transfer rates to and from NPC2 are very rapid, on the order of seconds or faster, this physical separation method serves only to provide an equilibrium distribution of the sterol between protein and membranes, but nevertheless indicates that the changes in NPC2 tryptophan fluorescence are reflecting sterol transfer between protein and membranes.

Although there is order of magnitude more rapid sterol transfer from NPC2 to LBPA-containing vesicles relative to EPC vesicles, no appreciable differences in the equilibrium distribution of cholesterol between NPC2 and the different membranes were found.

## LBPA restores normal NPC2 cholesterol transfer rates for proteins with mutations outside the hydrophobic knob

Using zwitterionic PC membranes we recently reported that point mutations in multiple regions on the NPC2 surface, including in the hydrophobic knob domain, led to diminished rates of cholesterol transfer between NPC2 and membranes (*McCauliff et al., 2015*); the impact of LBPA on cholesterol transfer by these NPC2 mutants was not investigated. Here we examined the rates of cholesterol transfer from these and additional point mutations in NPC2 protein to membranes containing 25 mol% LBPA. Mutations were confirmed by DNA sequencing and all mutant proteins were found to bind cholesterol similar to WT NPC2 (*Friedland et al., 2003*; *Ko et al., 2003*), with submicromolar affinity (*McCauliff et al., 2015*). The results in *Figure 3A* show that when acceptor membranes included 25 mol% of LBPA, cholesterol transfer rates for NPC2 proteins with mutations in regions *other* than the hydrophobic knob were similar to those of WT. Surprisingly, though mutations at H31, Q29, D113, and E108 exhibited sterol transfer rates to zwitterionic EPC membranes that were ≤15% of WT NPC2, the inclusion of LBPA in acceptor membranes resulted in rates of cholesterol transfer that were ≥85% of WT rates. By contrast the I62 and V64 mutations, both in the hydrophobic knob and which also resulted in markedly defective cholesterol transfer to EPC vesicles, were unaffected by the inclusion of LBPA in the acceptor membranes, with cholesterol transfer by these mutants remaining barely detectable. The G61A mutation, also in the hydrophobic knob, resulted in cholesterol transfer deficiencies similar to the I62 and V64 mutants, though changes are less extreme; sterol transfer to EPC vesicles was reduced by 70%, and it remained highly defective in the presence of LBPA, with rates of cholesterol transfer of only 16% relative to WT NPC2. Mutations in hydrophobic knob residues H56, G57, and I58 had little effect on cholesterol transfer rates to EPC vesicles, however unlike the WT NPC2, these mutants were insensitive to the presence of LBPA in acceptor membranes.

The NPC2 residues where mutations cause large decreases in cholesterol transfer rates to EPC are shown in red in *Figure 3B*; multiple surface regions are highlighted, in agreement with our aforementioned studies (*McCauliff et al., 2015*). In striking contrast, *Figure 3C* shows that the mutations which remained insensitive to membrane LBPA (shown in red) were localized exclusively in the hydrophobic knob domain; all other surface mutations, including many that were markedly defective in sterol transfer to EPC membranes, were sensitive to LBPA inclusion and displayed normalized rates of sterol transfer (shown in green). These results strongly indicate that the hydrophobic knob domain is the sole LBPA-sensitive region on the protein surface.

## Effects of NPC2 mutations on vesicle-vesicle interaction also highlight the hydrophobic knob domain

We and others have shown that WT NPC2 promotes membrane-membrane interactions (*Abdul-Hammed et al., 2010*; *Berzina et al., 2018*; *McCauliff et al., 2011*; *McCauliff et al., 2015*). We further showed that NPC2 point mutants with deficient cholesterol transfer abilities are also unable to cause EPC membrane aggregation (*McCauliff et al., 2015*). In the present studies, we investigated whether the presence of LBPA in the vesicles affected vesicle aggregation by WT and mutant NPC2 proteins. The results in *Table 1* show that inclusion of 25 mol% LBPA in LUVs resulted in a 16-fold increase in the rate of membrane-membrane interaction by WT NPC2, relative to 100% EPC LUVs. Incorporation of LBPA into membranes normalizes the membrane aggregation rates for NPC2 proteins with mutations outside the hydrophobic knob, for example H31, D113, and Q29. By contrast, the hydrophobic knob domain mutations were relatively insensitive to membrane LBPA. The results for this membrane-membrane interaction assay map virtually identically onto the NPC2 structure as did those for the cholesterol transport rates, as seen in *Figure 3C*.

**A.**

| | 100% EPC | | 25%LBA/EPC SUV | | |
|---|---|---|---|---|---|
| | Absolute Rate (s⁻¹) | Relative Rate | Absolute Rate (s⁻¹) | Relative to WT with LBPA | Relative to WT with EPC |
| WT | 0.0055 ± 0.0005 | 1.00 ± 0.09 | 0.0351 ± 0.0030 | 1.00 ± 0.08 | 6.38 ± 1.63 |
| E108A | 0.0018 ± 0.0007 | *0.32 ± 0.04* | 0.0404 ± 0.0011 | 1.15 ± 0.09 | 7.36 ± 0.62 |
| D72A | <0.0001 | *<0.01* | 0.0317 ± 0.0035 | 0.90 ± 0.10 | 5.77 ± 1.89 |
| H31A | 0.0007 ± 0.0001 | *0.13 ± 0.01* | 0.031 ± 0.02 | 0.89 ± 0.07 | 5.57 ± 0.37 |
| Q29A | 0.0006 ± 0.0001 | *0.11 ± 0.02* | 0.030 ± 0.019 | 0.85 ± 0.12 | 5.39 ± 0.83 |
| D113A | <0.0001 | *<0.01* | 0.040 ± 0.03 | 1.14 ± 0.13 | 7.21 ± 1.23 |
| I62D | <0.0001 | *<0.01* | <0.0001 | *<0.01* | *<0.01* |
| V64A | <0.0001 | *<0.01* | <0.0001 | *<0.01* | *<0.01* |
| H56A | 0.0055 ± 0.0001 | 0.99 ± 0.03 | 0.016 ± 0.0004 | *0.45 ± 0.05* | *2.97 ± 0.39* |
| G57D | 0.0058 ± 0.0001 | 1.07 ± 0.12 | 0.011 ± 0.003 | *0.31 ± 0.02* | *2.07 ± 0.18* |
| I58A | 0.0052 ± 0.0001 | 0.94 ± 0.02 | 0.0052 ± 0.0003 | *0.15 ± 0.06* | *0.93 ± 0.08* |
| G61A | 0.0017 ± 0.0004 | *0.31 ± 0.02* | 0.0055 ± 0.0004 | *0.16 ± 0.03* | *1.00 ± 0.21* |

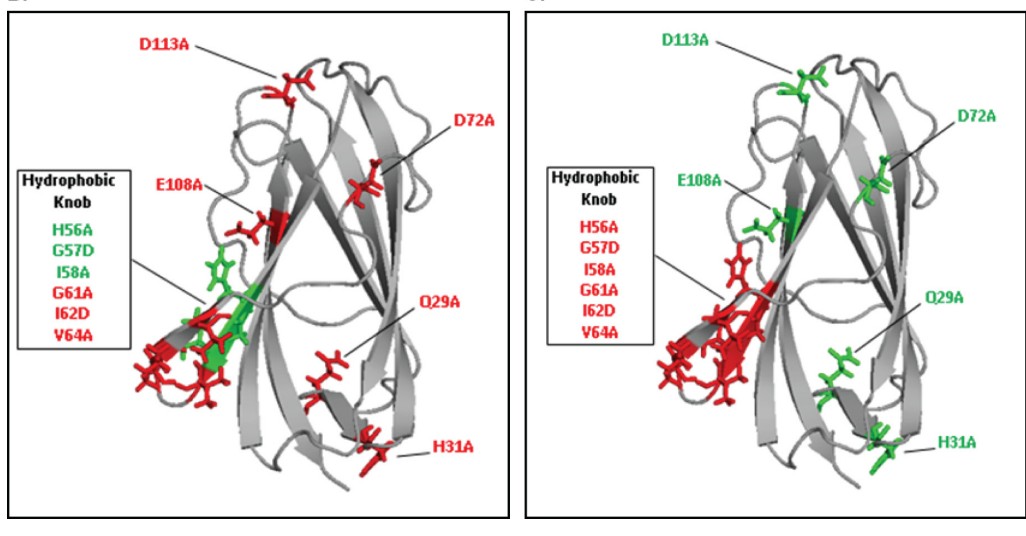

**Figure 3.** LBPA cannot reverse cholesterol transport deficiencies of NPC2 hydrophobic knob mutants. (**A**) Transfer of cholesterol from 1 µM WT or mutant NPC2 to 125 µM 100% EPC or 25% LBPA/EPC vesicles was measured on an SX20 Stopped Flow Spectrofluorometer by monitoring the dequenching of NPC2 endogenous tryptophan fluorescence. All curves were well fit using a single exponential function using the Applied Photophysics Pro-Data Viewer software. Mutants with rates of cholesterol transfer less than 50% of WT NPC2 were considered to have defective transfer kinetics properties and their relative rates are italicized. Data are representative of 3 experiments, each consisting of 2–3 individual runs. Absolute and relative rates of transfer to each model membrane system, ± SE, are shown. (**B**) NPC2 point mutations resulting in defective cholesterol transport to 100% EPC vesicles are shown in red while mutations having little or no effect on NPC2 cholesterol transport properties relative to WT protein, are shown in green. (**C**) Point mutations with attenuated rates of cholesterol transport to 25% LBPA/EPC vesicles are shown in red while mutations having little or no effect, relative to WT protein, are shown in green.

DOI: https://doi.org/10.7554/eLife.50832.005

## NPC2 interaction with LBPA and other phospholipids, and identification of the LBPA-interactive domain

To determine whether the relationship between LBPA and NPC2 in cholesterol trafficking involves direct interactions, protein-lipid binding assays were conducted. For studies of WT NPC2 protein, custom LBPA Snoopers (Avanti Polar Lipids) containing various LBPA isomers were incubated at pH 7.4 with WT NPC2 and relative binding was assessed via densitometric analysis of an antibody-probed strip, as described in Methods. The results shown in *Figure 4A* demonstrate that WT NPC2 binds to LBPA, showing greater interaction with isomers containing oleoyl (C18:1) as opposed to myristoyl (C14:0) fatty acyl chains, and overall the greatest degree of binding to the *S,S* 18:1 LBPA. NPC2 binding to *S,S* 18:1 LBPA was also greater than binding to egg PC.

**Table 1.** Rate of NPC2-mediated membrane interaction greatly increases in the presence of LBPA.

The effect of surface residue mutations on the ability of NPC2 to induce vesicle-vesicle interactions was assessed by measuring absorbance at 350 nm (light scattering) of 200 μM LUVs in the presence of 1 μM WT or mutant NPC2 protein, as described under Materials and methods. Rates of vesicle-vesicle interactions, indicated by increases in A350nm over time, were determined by a three-parameter hyperbolic fit of the data using Sigma Plot software, and are representative of at least three individual experiments. Mutants with substantially attenuated rates of membrane aggregation are indicated in italics.

| | 100% EPC | | 25%LBA/EPC SUV | | |
| --- | --- | --- | --- | --- | --- |
| | Absolute rate (s$^{1-}$) | Relative rate | Absolute rate (s$^{-1}$) | Relative to WT with LBPA | Relative to WT with EPC |
| WT | 0.0112 ± 0.0006 | 1.00 ± 0.05 | 0.1674 ± 0.0122 | 1.00 ± 0.07 | 14.95 ± 1.09 |
| H31A | 0.0018 ± 0.0005 | *0.16 ± 0.05* | 0.1262 ± 0.0079 | 0.75 ± 0.05 | 11.27 ± 0.71 |
| D113A | <0.0001 | *<0.01* | 0.1573 ± 0.0112 | 0.94 ± 0.07 | 14.04 ± 1.00 |
| Q29A | 0.0029 ± 0.0008 | *0.26 ± 0.07* | 0.1337 ± 0.0039 | 0.80 ± 0.02 | 11.93 ± 0.35 |
| E108A | 0.0032 ± 0.0004 | *0.29 ± 0.03* | 0.1437 ± 0.0094 | 0.86 ± 0.06 | 12.83 ± 0.84 |
| D72A | 0.0032 ± 0.0003 | *0.29 ± 0.03* | 0.1456 ± 0.0077 | 0.87 ± 0.05 | 13.00 ± 0.69 |
| H56A | 0.0107 ± 0.0008 | 0.95 ± 0.07 | 0.0261 ± 0.0055 | *0.16 ± 0.09* | *2.33 ± 0.49* |
| G57D | 0.0103 ± 0.0016 | 0.92 ± 0.14 | 0.0202 ± 0.0066 | *0.13 ± 0.12* | *1.81 ± 0.59* |
| I58A | 0.0094 ± 0.0011 | 0.840 ± 0.10 | 0.0240 ± 0.0037 | *0.15 ± 0.06* | *2.14 ± 0.33* |
| G61A | 0.0075 ± 0.0006 | *0.67 ± 0.06* | 0.0406 ± 0.0012 | *0.25 ± 0.03* | *3.63 ± 0.11* |
| I62D | <0.0001 | *<0.01* | 0.0167 ± 0.0017 | *0.10 ± 0.03* | *1.50 ± 0.15* |
| V64A | <0.0001 | *<0.01* | 0.0193 ± 0.0040 | *0.12 ± 0.06* | *1.72 ± 0.36* |

DOI: https://doi.org/10.7554/eLife.50832.006

Binding of WT NPC2 to other typical membrane phospholipid species, in comparison to di-oleoyl LBPA, was also examined at pH 7.5 using membranes spotted with di-oleoyl phospholipids. *Figure 4B* shows that NPC2 interacts more strongly with LBPA than with PC, PA, PG, and PS species. As two other anionic phospholipids assayed, PG and PA, show even less interaction with NPC2 than does zwitterionic PC, the mechanism by which NPC2 binds to LBPA is likely not solely dependent on electrostatic interactions.

In protein-lipid overlay assays the phospholipids are not necessarily present in a physiological orientation, therefore we further examined NPC2–lipid interaction using Homogenous Time Resolved Fluorescence (HTRF), in which the phospholipids are present as lamellar structures. HTRF, which has been recently demonstrated to be an effective and sensitive assay for lipid-protein interaction (*Fleury et al., 2015*), is a fluorescence resonance energy transfer (FRET) based technology that utilizes specific fluorophores that emit long-lived fluorescence signals when involved in a FRET process. Notably the europium cryptate energy donor has been shown to be impervious to photobleaching and to exhibit significant stability in a homogenous assay environment (*Degorce et al., 2009*). This allows for time-resolved measurements of, in this case, protein-lipid interactions, with the ability to easily subtract background signals. The NPC2-interaction with LBPA at pH 5.0 was substantially greater than with other negatively charged lipids or with zwitterionic PC (*Figure 4D*), in general agreement with the lipid-blot results. Taken together the results support direct interactions between NPC2 and membrane phospholipids, the greatest interaction being with LBPA.

To determine whether there is a specific LBPA interactive domain on the surface of NPC2, we further employed the LBPA lipid blots to examine interactions with NPC2 point mutants. The results show that mutations in most of the hydrophobic knob residues markedly reduce interactions, relative to WT NPC2. For example, studies analyzing binding to various isomers of LBPA show that the hydrophobic knob mutants I62N and V64A bound the *S,S* and *S,R* di-oleoyl isomers at only ~20% of WT levels, and the C18:1 *R,R* isomer at approximately 40% of WT. In contrast, the Q29A and D113A mutants, in regions outside the hydrophobic knob, bound nearly all LBPA isomers similar to WT; only Q29A was observed to bind the C18:1 *R,R* isomer at approximately 50% relative to WT binding (*Figure 5A*). The I62D and V64A mutants exhibited greater interaction with the 18:1 Semi LBPA species, which has three oleoyl acyl chains (*Figure 5A*), binding approximately 70% relative WT NPC2. Mutations in other hydrophobic knob residues also reduced the interaction of NPC2 with the di-

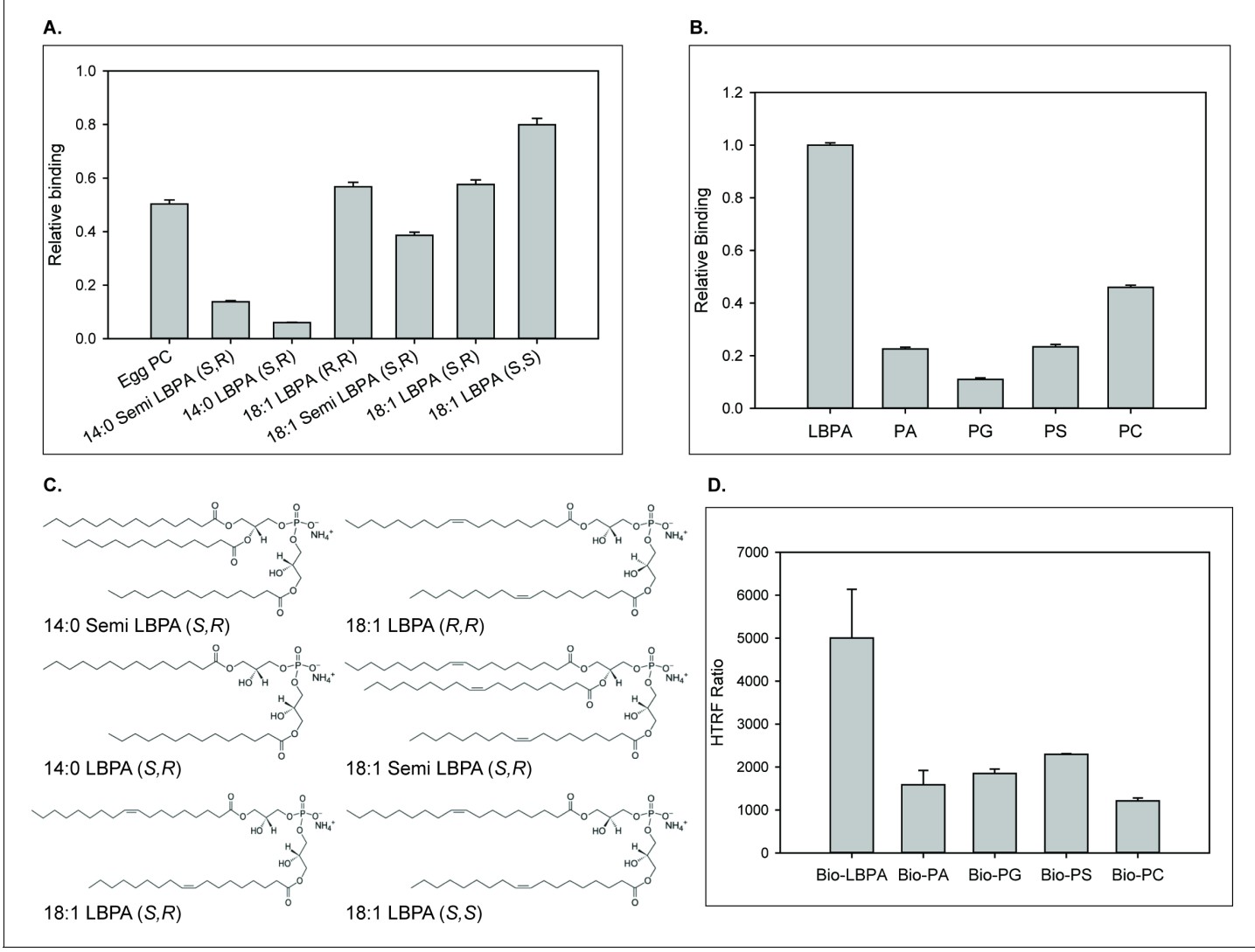

**Figure 4.** NPC2 binding to LBPA and other phospholipids. (**A**) WT NPC2 protein was incubated with strips (Snoopers) containing LBPA isomers. LBPA-bound protein was detected with anti-c-myc antibody as described under Methods, and degree of binding ± SE (n = 3) is represented by the integrated density of the blots. (**B**) 500 pmol of various membrane phospholipids were spotted onto nitrocellulose strips and probed with WT NPC2-myc-his protein as described under Methods. Relative binding of WT NPC2 ± SE (n = 5) is shown, represented by signal intensity detected with the LI-COR system. (**C**) Structures of the LBPA isomers. (**D**) 75 nM of WT NPC2 protein was incubated with 1 μM of the indicated biotin-C12-ether phospholipid, streptavidin-d2 conjugate and europium cryptate–labeled monoclonal anti-histidine antibody in detection buffer, as described in Methods. FRET signal between europium cryptate and streptavidin was detected with a HTRF capable Envision plate reader (λex = 320 nm, λem = 615 and 665 nm; 100 μs delay time; n = 3).

DOI: https://doi.org/10.7554/eLife.50832.007

oleyol *S,S* LBPA on blots using various phospholipids. Indeed, similar to what was observed with the LBPA isomer blots, the H56, G57, I58, and G61 mutants showed only 30% to 40% degree of interaction relative WT NPC2. In marked contrast, surface mutations outside the hydrophobic knob had virtually no impact on NPC2 interaction with LBPA, with Q2A, H31A, D113, and E108 mutants exhibiting WT levels of interaction with LBPA (*Figure 5B*). Overall, mutations within the hydrophobic knob domain of NPC2 resulted in diminished binding of the protein to LBPA while mutations outside the knob region presented proteins with LBPA interactions similar to WT (*Figure 5B*). HTRF analysis of the mutant NPC2 proteins was generally consistent with the lipid blot results, also indicating reduced binding of the hydrophobic knob mutants to LBPA (*Figure 5C and D*).

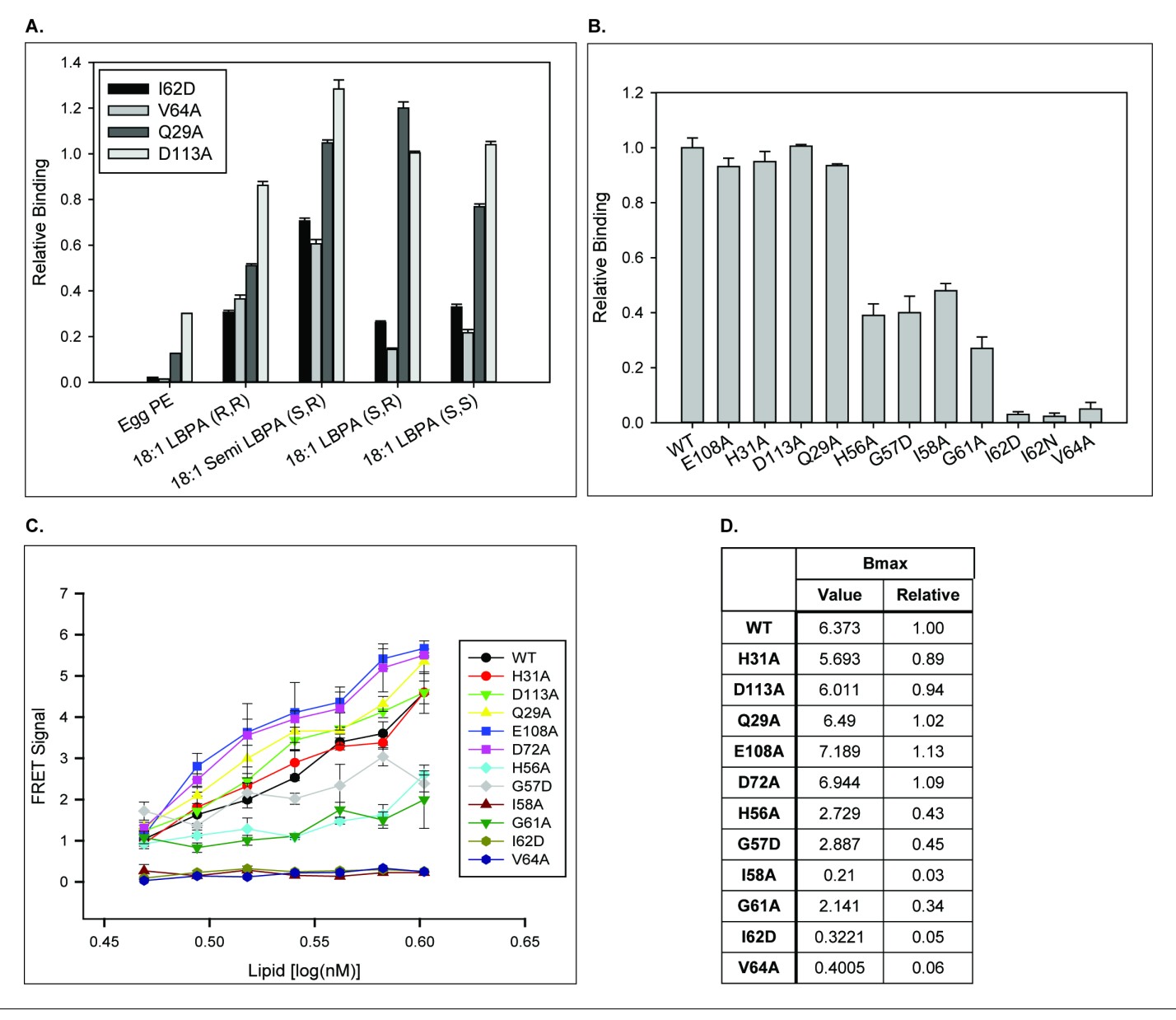

**Figure 5.** NPC2 binds to LBPA via the hydrophobic-knob domain. Binding of NPC2 WT and mutant proteins to (**A**) LBPA isomers and (**B**), 18:1 (S,R) LBPA was detected using LBPA Snoopers and membranes spotted with 500 pmol phospholipid, respectively, as described under Methods. Relative binding is represented as (**A**) the integrated density of the blots, relative to WT NPC2, ± SE (n = 3) and (**B**) the signal intensity detected with the LI-COR system ± SE (n = 3) (**C**). HTRF analysis: WT or mutant NPC2 protein was incubated with biotin-C12-ether LBPA, streptavidin-d2 conjugate and europium cryptate–labeled anti-His antibody in detection buffer, as described in Methods. FRET signal between europium cryptate and streptavidin was detected with a HTRF capable Envision late reader as described in Methods. (**D**) FRET signal was analyzed with the One site–specific binding function (Graphpad) and the Bmax extrapolations were used to infer binding capacity between recombinant NPC2 protein and biotin-C12-ether LBPA. Results are representative of three experiments, with deviations $\leq$ 20%.

DOI: https://doi.org/10.7554/eLife.50832.008

## Transfer kinetics and LBPA interaction predict the ability of WT or mutant NPC2 to reduce cholesterol accumulation in NPC2 deficient fibroblasts

In agreement with several previous reports, incubation of NPC2 deficient fibroblasts with WT NPC2 protein resulted in a dramatic decrease in filipin staining, reaching levels similar to healthy fibroblasts

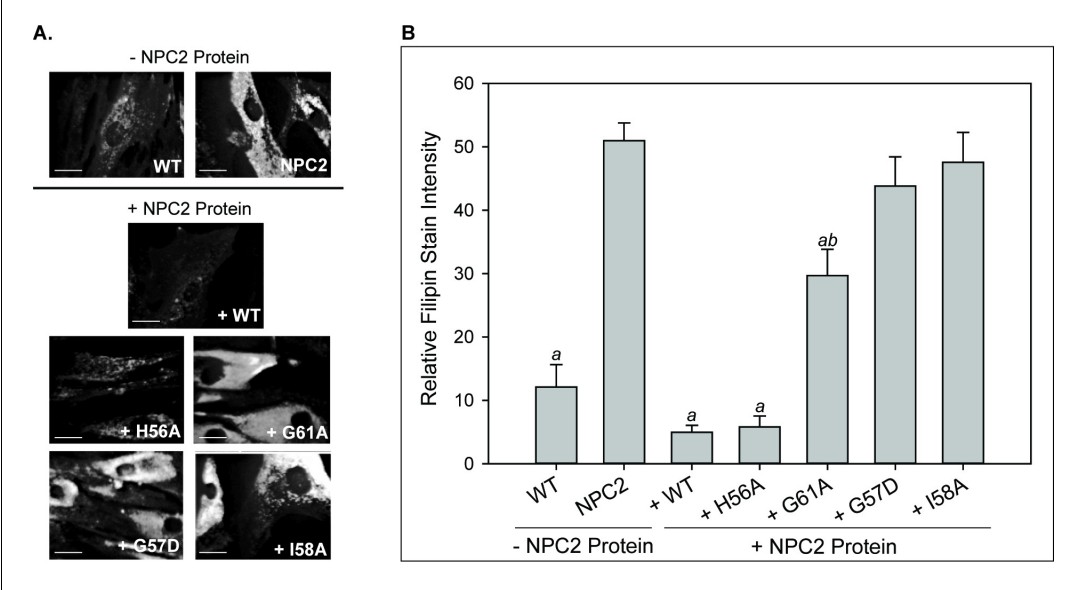

**Figure 6.** LBPA insensitive mutants are unable to rescue cholesterol accumulation in NPC2 patient fibroblasts. NPC2-deficient fibroblasts were incubated with 0.4 nm purified WT or NPC2 mutant protein and cholesterol accumulation was quantified via filipin staining as described in the Methods. (A) Representative microscopy images of filipin stained control and treated NPC2-deficient fibroblasts. Scale bars, 70 μM. (B) Filipin stain intensity of untreated control and treated NPC2-deficient fibroblasts was quantified and are presented as percent of control. Data are representative of at least three separate experiments ± SE. a, p<0.01 vs untreated NPC2-deficient cells; b, p<0.01 vs untreated WT cells by Student's t-test.
DOI: https://doi.org/10.7554/eLife.50832.009

(*Figure 6*) (*Ko et al., 2003*; *Liou et al., 2006*; *McCauliff et al., 2011*; *McCauliff et al., 2015*). The H56A mutant, with rates of sterol transfer and membrane aggregation similar to WT protein, also reduced filipin stain area, similar to WT NPC2. In contrast, the G57D and I58A hydrophobic knob mutants, with markedly attenuated cholesterol transfer and membrane aggregation rates, were unable to reverse cholesterol accumulation in NPC2 cells; filipin staining remained at a level comparable to that of the unsupplemented cells. G61A, also in the knob domain, was able to lessen cholesterol accumulation in NPC2 cells to a moderate extent, and its defect in cholesterol transfer was also more modest than that of other hydrophobic knob mutants (*Figure 6*). These results are in keeping with our previously reported results for two other hydrophobic knob mutants, I62D and V64A (*McCauliff et al., 2015*) which are also deficient in cholesterol transfer and membrane aggregation ability. The consistency between results of the sterol transfer assays and membrane-membrane interaction assays, with the cholesterol clearance in patient cells, strongly supports the physiological relevance of the structure-function studies, and points to a particularly important role for the hydrophobic knob of NPC2 in effecting normal sterol trafficking.

## PG supplementation increases LBPA levels in WT, NPC1- and NPC2-deficient cells

To increase cell LBPA levels, NPC patient fibroblasts were incubated with PG, known to be its precursor (*Bouvier et al., 2009*; *Poorthuis and Hostetler, 1978*; *Thornburg et al., 1991*). The results in *Figure 7* show that incubation of the cells with 100% PG SUVs led to substantial increases in cellular content of LBPA in all fibroblast types; in WT cells the increase was nearly 6-fold. In agreement with previous reports LBPA levels in NPC patient cells were found to be increased relative to WT cells prior to enrichment (*Chevallier et al., 2008*; *Davidson et al., 2009*; *Sleat et al., 2004*; *Vanier, 1983*); PG incubation resulted in 2–3 fold increases in NPC1- and NPC2-deficient cells, respectively, relative to unsupplemented cells (*Figure 7A*). Levels of other cellular phospholipids appeared unchanged (*Figure 7B*). Direct addition of LBPA-containing SUVs also led to approximately 2 to 3-fold increases in cellular LBPA levels (data not shown), in agreement with

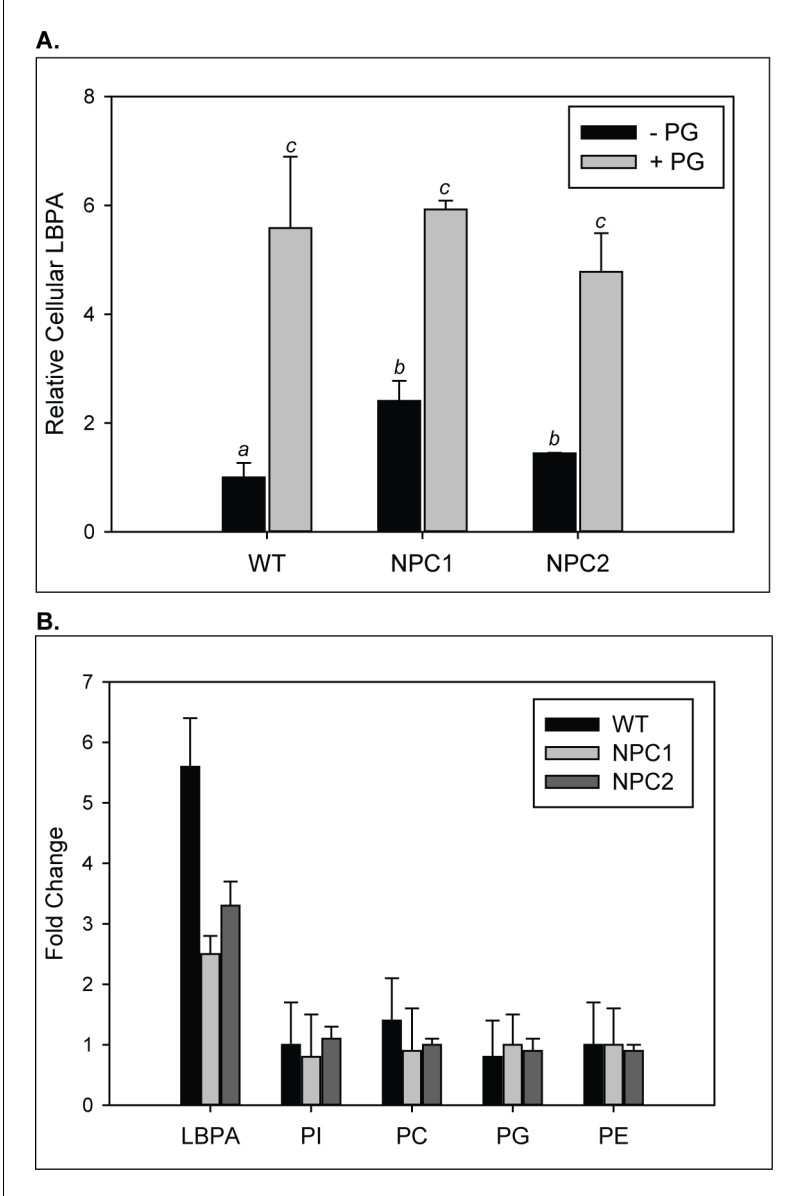

**Figure 7.** Increase in LBPA content in cells supplemented with PG. WT, NPC1-, and NPC2 deficient fibroblasts were incubated with 100 μM PG SUVs. (**A–B**) Lipids were extracted and phospholipids quantified by TLC as described under Methods. Results are representative of two experiments, each conducted in duplicate, ± SE. (**A**) Data are normalized to LBPA levels in untreated WT cells. (**B**) Fold changes in PL species in PG-treated relative to untreated cells; *p<0.01 between PL species by one-way ANOVA.
DOI: https://doi.org/10.7554/eLife.50832.010

(**Chevallier et al., 2008**). Supplementation with PC SUVs as a control had no effect on the phospholipid composition of any of the cell types (data not shown).

## PG supplementation/LBPA enrichment reduces cholesterol accumulation in NPC1- but not NPC2-deficient fibroblasts

Following PG supplementation, cholesterol content remained unchanged in the WT fibroblasts (**Figure 8**). NPC1 deficient fibroblasts exhibited a dramatic reduction in cholesterol accumulation following PG supplementation, approaching levels observed for WT cells and similar to the direct addition of LBPA (**Chevallier et al., 2008**). In marked contrast to NPC1 deficient cells, the cholesterol

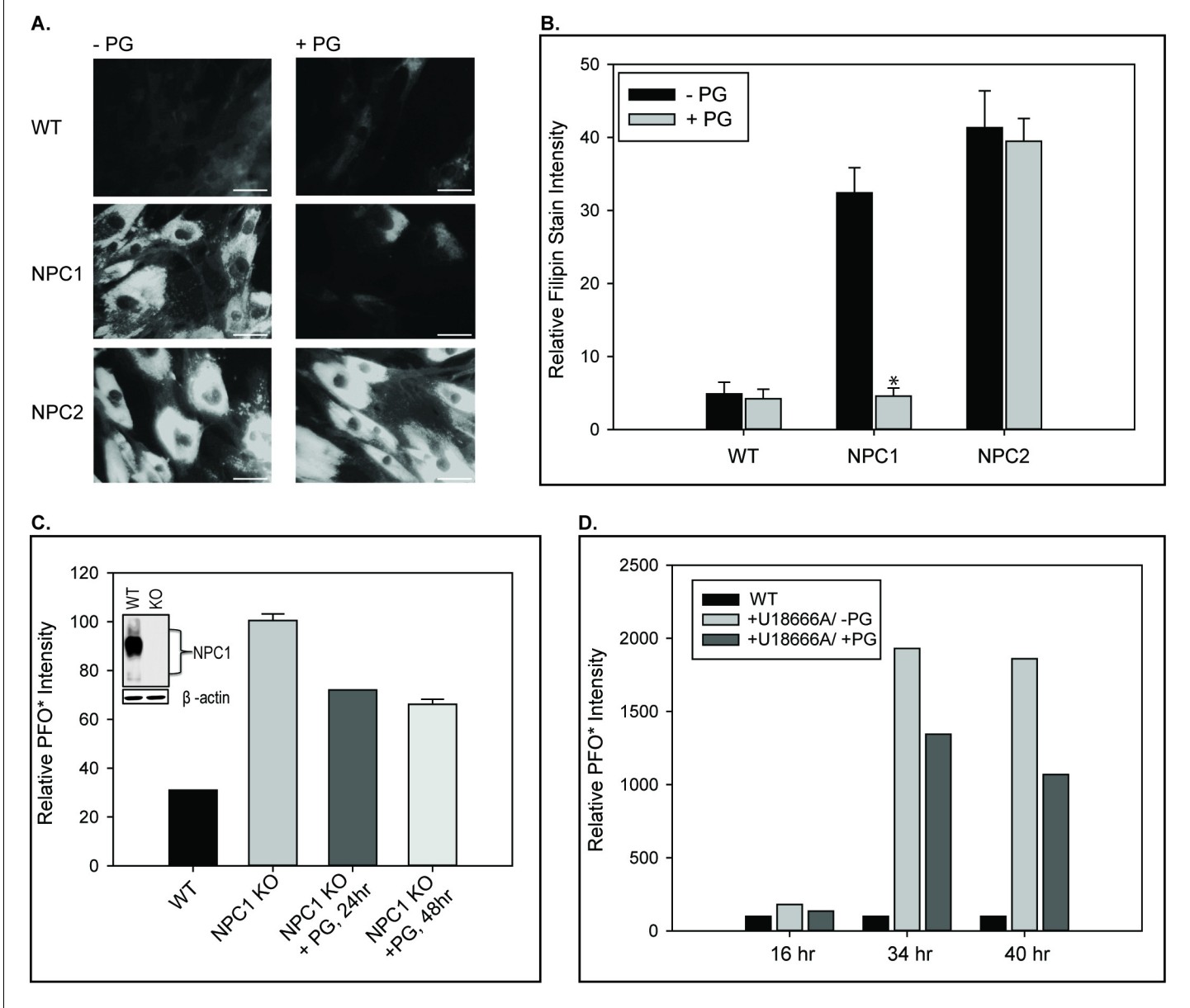

**Figure 8.** PG supplementation reverses cholesterol accumulation in NPC1– but not NPC2–deficient cells. WT, NPC1–, and NPC2– deficient fibroblasts were incubated with 100 µM PG SUVs and cholesterol accumulation was determined by filipin staining as described under Methods. (**A**) Representative images of untreated and PG supplemented fibroblasts stained with filipin. Scale bars, 70 µM. (**B**) Percent filipin stain intensity, relative to control condition. Data are representative of three individual experiments, ± SE. *p<0.01 vs untreated cells by Student's *t*-test. The effect of PG supplementation on cholesterol accumulation in NPC1-deficiency was also measured by PFO* intensity, assessed via flow cytometry, in (**C**) CRISPR-Cas9 mediated NPC1 knockout HeLa cells and (**D**) WT human fibroblasts treated with 1 µM U-18666A. Data are expressed relative to (**C**) untreated NPC1 knockouts and (**D**) untreated WT fibroblasts per incubation period.

DOI: https://doi.org/10.7554/eLife.50832.011

accumulation in NPC2 deficient fibroblast remained elevated following PG supplementation, despite increased LBPA content (*Figure 8*).

While the PG/LBPA-dependent cholesterol egress observed in NPC1 deficient cells appears to indicate an NPC1-independent mechanism for cholesterol egress, the NPC1 cell line used in these studies is a compound heterozygote (C709T and T3182C) which, despite its well established deficiency in cholesterol egress, is still expressed in the limiting LE/LY membrane and thus could potentially be involved in the egress of cholesterol from cells supplemented with PG. To address this

possibility, we generated NPC1 null HeLa cells via CRISPR-Cas9 and found that PG supplementation led to significant reductions in the cholesterol accumulation phenotype (*Figure 8C*). We additionally performed PG supplementation in WT cells treated with the U18666A compound, known to induce the NPC disease phenotype at the cellular level by targeting NPC1 (*Lu et al., 2015*). In these cells too, PG supplementation resulted in cholesterol clearance (*Figure 8D*). Overall, these results suggest that LBPA can promote cholesterol egress from the endolysosomal compartment despite the absence of NPC1, but not in the absence of NPC2.

## Cellular LBPA enrichment restores cholesterol clearance properties of LBPA-sensitive NPC2 mutants

The point mutagenesis analysis of NPC2 shows a direct relationship between the cholesterol transfer rate of a particular NPC2 mutant and its ability to rescue the cholesterol accumulation of NPC2-deficient cells (*McCauliff et al., 2015*; *Figures 3* and *6*). In the present studies we discovered an unanticipated impact of LBPA incorporation into membranes, in which some NPC2 mutants that were highly defective in sterol transfer to phosphatidylcholine membranes, were essentially normalized when LBPA was present. These 'LBPA-sensitive' NPC2 mutations were, almost exclusively, present in surface domains outside of the hydrophobic knob. NPC2 proteins with mutations in the hydrophobic knob, that were highly defective in cholesterol transfer to phosphatidylcholine membranes, remained insensitive to LBPA in the membranes (*Figure 3*). Based on this structure-based difference in LBPA sensitivity, we hypothesized that increasing the LBPA levels in NPC2 deficient cells would enhance the activity of mutants outside the hydrobphobic knob that responded to LBPA in the kinetic assays, whereas cellular LBPA enrichment would not enhance the action of the hydrophobic knob mutants that were insensitive to LBPA. To test these predictions, NPC2 deficient fibroblasts were incubated with purified wild type or mutant NPC2 proteins, with or without PG supplementation, and cholesterol accumulation was assessed by filipin staining as described. The results in *Figure 9* demonstrate that NPC2 proteins with mutations outside the hydrophobic knob, such as Q29A, D113A, and D72A, which were unable to reduce cholesterol accumulation in unsupplemented NPC2 deficient cells, were indeed able to 'rescue' cells that were enriched with LBPA via PG supplementation. By contrast, the hydrophobic knob mutants I62D and G61A, which were insensitive to LBPA in cholesterol transfer assays, were unable to clear cholesterol from LBPA-enriched NPC2 deficient cells. Thus, the results show that a combination of increased cellular levels of LBPA and LBPA-sensitive NPC2 protein was able to rescue the NPC2 deficient cells, beyond the ability of the mutant protein alone (*Figure 9*). As before, WT cells supplemented with PG alone showed no change in cholesterol accumulation, and cells supplemented with purified WT NPC2, with or without PG, showed significantly reduced filipin staining. The results in patient cells again mirror the results of sterol kinetics experiments, suggesting that return to normal sterol trafficking can be achieved for NPC2 mutations outside the hydrophobic knob that are otherwise dysfunctional, when the LBPA content of the cells is increased.

## Discussion

LBPA was proposed to be involved in intracellular cholesterol trafficking based on the sterol accumulation which accompanies incubation of cells with an anti-LBPA antibody (*Kobayashi et al., 1999*), however the mechanism of LBPA action has remained unknown. In this study we have, for the first time, shown that LBPA function in cholesterol trafficking is obligately dependent upon its interaction with the NPC2 protein within the endo-lysosomal system. At the molecular level, we show that this novel step in intracellular cholesterol trafficking involves direct interaction of LBPA with the hydrophobic knob domain on NPC2. This surface domain is located near the cholesterol binding pocket, thus its insertion into the bilayer would position the protein to efficiently exchange cholesterol with the membrane. Recent molecular dynamic simulations show that LBPA, but not other phospholipids, may position NPC2 in an orientation that could promote protein-membrane sterol exchange (*Enkavi et al., 2017*), potentially inducing a conformational change in the binding pocket of NPC2 such that the off-rate of cholesterol is markedly increased, allowing for rapid transfer of cholesterol between protein and LBPA-enriched membranes.

A decade ago, the Gruenberg laboratory reported that viral-mediated supplementation of NPC1-deficient cells with exogenous LBPA reversed cholesterol accumulation in the diseased cells

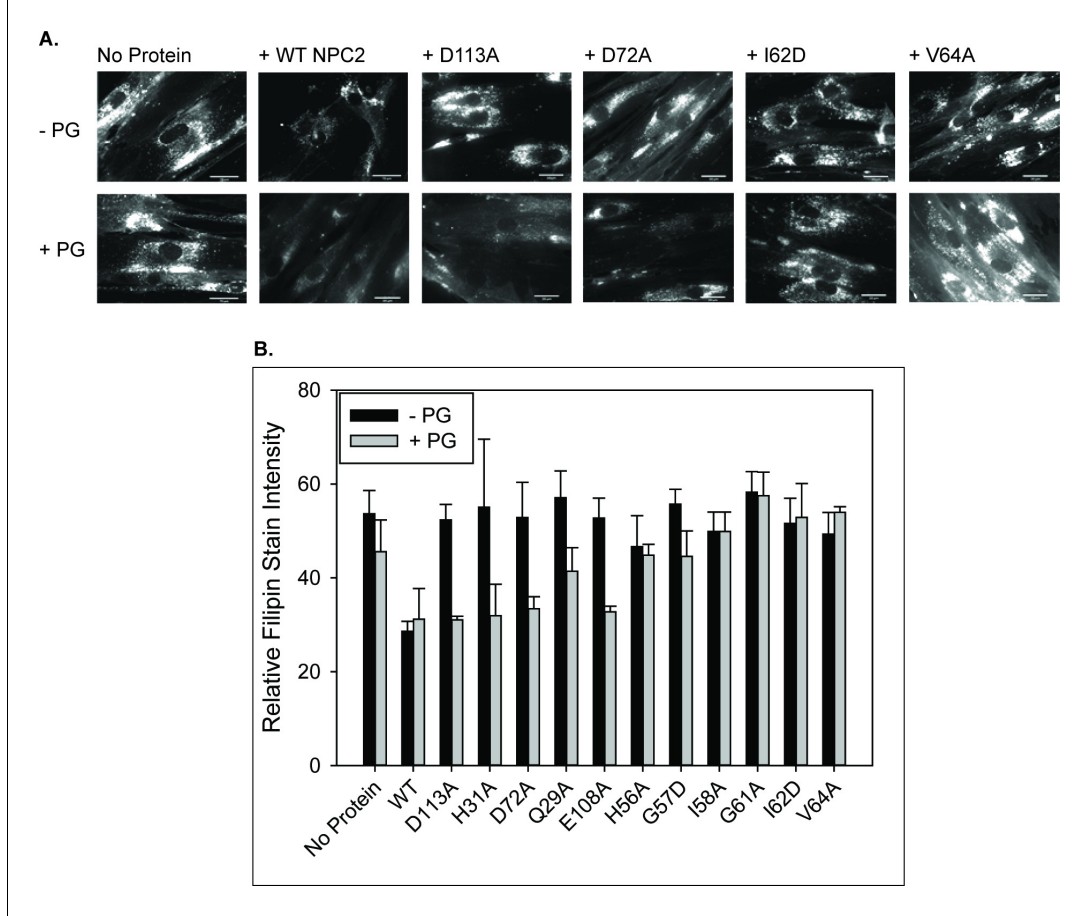

**Figure 9.** LBPA–sensitive but not insensitive NPC2 mutants reverse cholesterol accumulation in NPC2–deficient cells when co-treated with PG. NPC2 deficient fibroblasts were incubated with purified WT or mutant NPC2 proteins alone or in the presence of PG SUVs, as described under Methods, and cholesterol accumulation was quantified by filipin staining. (**A**) Representative images of treated NPC2 deficient fibroblasts stained with filipin. Scale bars, 70 μM. (**B**) Data are from four or more individual incubations ± SE, *p<0.01 vs no PG supplementation by Student's *t*-test.
DOI: https://doi.org/10.7554/eLife.50832.012

(*Chevallier et al., 2008*). Here we show that LBPA enrichment is, conversely, completely ineffective in cells expressing NPC1 but lacking NPC2, underscoring the required functional interaction of LBPA with the NPC2 protein. In our efforts to establish the molecular basis of this interaction, we found that several NPC2 mutants that could not reverse cholesterol accumulation in unsupplemented NPC2-deficient cells, became effective if the cells were first enriched with LBPA. The NPC2 mutations which were 'rescued' by LBPA enrichment are all localized outside of the hydrophobic knob domain. By contrast, mutations within the hydrophobic knob were insensitive to LBPA enrichment apart from H56A, which may be on the limiting edge of this region. Thus, LBPA is able to normalize deficient transfer by mutant NPC2 as long as the mutation is outside of the hydrophobic knob. Interestingly, direct interaction between NPC2 and NPC1, residing in the limiting LE/LY membrane, has also been identified as involving the hydrophobic knob region on NPC2 (*Li et al., 2016*; *Wang et al., 2010*). This highlights further the critical nature of this domain in intracellular cholesterol trafficking, such that the NPC2 hydrophobic knob is involved in both LBPA-dependent binding of cholesterol by NPC2 at inner-LE/LY membranes, and the delivery of cholesterol by NPC2 to the NPC1 N-terminal domain. Primary amino acid sequence comparisons reveal a high degree of conservation within the NPC2 hydrophobic knob domain for all species with the exception of yeast; interestingly, yeast lacks the LBPA phospholipid, indirectly supporting a functional interaction between the mammalian NPC2 hydrophobic knob and LBPA. Thus, we propose that the hydrophobic knob on NPC2 interacts with LBPA in the inner LE/LY membranes, leading to cholesterol extraction, with

NPC2 then subsequently involved in the 'handoff' of the sterol to NPC1 in the limiting membrane of these organelles (*Deffieu and Pfeffer, 2011*; *Infante et al., 2008*; *Li et al., 2016*; *Wang et al., 2010*).

In healthy cells, the enrichment of inner LE/LY membranes with LBPA and concurrent decline in cholesterol content may be reflective of the efficient role LBPA plays in normal LE/LY cholesterol efflux. If direct interactions between LBPA and NPC2 are integral to this mechanism of efflux, a rate determining step may be their frequency of interaction and, thus, be partially dependent upon the concentration of both phospholipid and protein in the LE/LY compartment. While the concentration of LE/LY LBPA has been shown to increase in parallel with cholesterol and other lipids in NPC disease (*Davidson et al., 2009*; *Sleat et al., 2004*; *Vanier, 1983*), as also found here, it is possible that LBPA levels nevertheless remain too low, as previously suggested (*Chevallier et al., 2008*), to provide support for NPC2-mediated transport of the elevated cholesterol load. Our results using PG supplementation to augment LBPA levels further indicate that LBPA may become limiting in NPC1 disease. Importantly, we show here that LBPA enrichment cannot reverse cholesterol accumulation caused by NPC2 deficiency unless the mutation is outside the hydrophobic knob domain in which case PG supplementation is effective, as it is in NPC1-deficient cells. Interestingly, Chen et al. observed a pronounced association of the NPC2 protein with inner LE/LY membranes in cells of *Npc1^{-/-}* mice (Balb/c *Npc1^{nih}*), relative to WT (*Chen et al., 2005*), suggesting that increased recruitment of NPC2 to inner LE/LY membranes in NPC1 disease, where LBPA concentration is high, may be a compensatory mechanism to alleviate sterol accumulation.

Beyond its obligate interaction with NPC2 within the LE/LY, it is not yet clear how increased LBPA leads to the efflux of LE/LY cholesterol in the absence of NPC1 protein, although in recent studies we have found that increased concentrations of cellular LBPA increase macroautophagy in NPC1 deficient cells (data not shown). It is well known that impaired autophagy is present in NPC disease (*Liao et al., 2007*; *Lieberman et al., 2012*; *Seranova et al., 2017*), and stabilization of this pathway has been shown to reduce cholesterol accumulation in NPC1 disease (*Dai et al., 2017*; *Ordonez et al., 2012*; *Sarkar et al., 2013*). LBPA also displays a variety of other unique functions within the endo/lysosomal system that could potentially promote cholesterol efflux. Studies have demonstrated, for instance, that its transient interaction with the ESCRT-associated protein, ALIX, is required for the proper formation of multivesicular structures within the LE/LY (*Matsuo et al., 2004*), where cholesterol is localized (*Fivaz, 2002*); it is also found, along with cholesterol, on exosomes derived from fusion of inner lamellae with the limiting LE/LY membrane (*Matsuo et al., 2004*). Furthermore, it has recently been proposed that LBPA may influence cholesterol homeostasis beyond the confines of late endosomes and lysosomes, having been shown to be necessary for lipid droplet formation via Wnt signaling within the endoplasmic reticulum, where cholesteryl esters are synthesized (*Scott et al., 2015*). Within the endo/lysosomal system, however, our results demonstrating that cellular enrichment of LBPA cannot reverse cholesterol accumulation in cells lacking functional NPC2 protein indicate that the cholesterol efflux mechanism utilized by LBPA is dependent upon its interaction with NPC2.

LBPA is reported to comprise only 1% of total cellular phospholipids, but about 15 mol% of total phospholipids in the LE/LY (*Chevallier et al., 2000*; *Kobayashi et al., 1998*; *Kobayashi et al., 2002*). There is no agreement regarding which isomeric form of LBPA is predominant in cells, as the fatty acids can be linked at the sn-2 or sn-3 positions of each glycerol in the *S* or *R* conformation, though the sn-2, sn-2' positions are currently favored (*Chevallier et al., 2000*; *Kobayashi et al., 1998*; *Kobayashi et al., 2002*; *Mason et al., 1972*; *Matsuo et al., 2004*). The 2,2'-LBPA was shown to be quite effective at mobilizing cholesterol in NPC1 disease cells while the 3,3'- and semi-LBPA isoforms were unable to promote sterol efflux. No difference in efficacy between *S,S*, *S,R*, and *R,R* LBPA isomers were noted, however, suggesting that the conformation of LBPA has no bearing on its ability to reverse cholesterol accumulation in NPC1 deficient cells (*Chevallier et al., 2008*). In agreement with this observation, we previously showed that the *S,S*, *S,R*, and *R,R* configurations of LBPA had little to no effect on in vitro cholesterol transfer rates by NPC2 protein (*Xu et al., 2008*). Similarly, in the present studies we observed little variation in NPC2 binding to these different LBPA stereoisomers. Regardless, we chose to use the presumed precursor and structural isomer of LBPA, phosphatidylglycerol, to increase the cellular LBPA content in these studies, obviating any potential concern about the LBPA isoform as the cells presumably generate the physiologically accurate form. PG is thought to convert to LBPA along the endo/lysosomal system (*Hullin-Matsuda et al., 2007*;

*Poorthuis and Hostetler, 1978*), and studies have demonstrated that exogenously administered PG can be converted to LBPA in vivo (*Somerharju and Renkonen, 1980*), although the anabolic pathway of this conversion remains unknown. The present demonstration that PG supplementation specifically increases the LBPA content of all cells tested supports the hypothesis that PG is a precursor to LBPA. Cellular conversion of PG to LBPA has also been demonstrated in mammalian alveolar macrophages (*Waite et al., 1987*), lymphoblasts (*Hullin-Matsuda et al., 2007*) and RAW macrophages (*Bouvier et al., 2009*).

In addition to variations in stereochemistry, the cellular fatty acyl chain components of LBPA have been found to vary (*Bouvier et al., 2009*), with oleic acid and docosahexaenoic acid (DHA) reported to be selectively incorporated (*Besson et al., 2006*; *Luquain et al., 2001*). Here we found acyl chain-dependent differences in NPC2-LBPA interactions, with reduced binding to the 14-carbon saturated dimyristoyl-LBPA species relative to the 18-carbon monounsaturated dioleoyl-LBPA species. In prior work we showed that cholesterol transfer from NPC2 to membranes containing dioleoyl-LBPA was >2 fold faster than transfer to vesicles with dimyristoyl-LBPA (*Xu et al., 2008*). Taken together, the results suggest that the acyl composition but not the steroconfiguration of LBPA may be important in normal LE/LY cholesterol efflux. Ongoing studies focused on identifying the specific acyl-chain species that are most effective at stimulating LE/LY cholesterol clearance in an NPC phenotype, for instance, should inform both our understanding of intracellular cholesterol transport and the development of therapies for lysosomal storage disorders.

Based on the present results, we propose that the NPC2 hydrophobic knob domain inserts into LBPA enriched inner LE/LY membranes, interacting directly with the phospholipid. Given the ability of NPC2 to promote membrane-membrane interactions (*Abdul-Hammed et al., 2010*; *McCauliff et al., 2011*; *McCauliff et al., 2015*), we speculate that this interaction between LBPA and NPC2 is involved in the formation of membrane contact sites which could potentially exist between closely apposed inner LE/LY membranes and facilitate rapid transfer of sterol. Membrane contact sites have been shown to be important in intermembrane lipid transfer (*Helle et al., 2013*; *Holthuis and Levine, 2005*; *Prinz, 2014*), although none have yet been specifically described within the multivesicular LE/LY. Our results show that NPC2 promotes membrane-membrane interaction, and further indicate the ability of NPC2 to bind to LBPA; this interaction may represent one membrane contact point on inner LE/LY membranes. Our previous kinetic studies suggested that NPC2 interacts with other membrane phospholipids as well, and preliminary molecular dynamics simulations indicate that unlike LBPA, where NPC2 interacts at the hydrophobic knob, these interactions occur at NPC2 surface sites other than the knob domain (data not shown); such interactions could represent a second contact point within the inner LE/LY membranes. Interestingly, it has now been demonstrated that NPC2 also binds directly to NPC1 which resides in the limiting LE/LY membrane, and possibly also with LAMP proteins in these same membranes (*Li et al., 2016*). These could also be potential tether points for the NPC2 and would effectively bring inner LE/LY cholesterol-laden membranes in closer proximity to the limiting LE/LY membrane, which cholesterol must ultimately cross to exit the compartment.

The key finding of the present studies is that the primary mechanism by which LBPA stimulates LE/LY cholesterol efflux is critically dependent upon its interaction with functional NPC2. LBPA first appears on the internal vesicles and membranes of cholesterol rich multivesicular bodies characteristic of late endosomes (*Gruenberg, 2003*; *Möbius et al., 2003*) and increases in concentration through the pathway to cholesterol depleted lysosomes (*Möbius et al., 2003*). This localization overlaps with NPC2, targeted to late endosomes/lysosomes via the mannose-6-phosphate receptor (*Naureckiene et al., 2000*). While NPC1 is also targeted to late endosomes (*Garver et al., 2000*; *Higgins et al., 1999*; *Neufeld et al., 1999*), Blanchette-Mackie and colleagues observed that most NPC1 localizes to a specific set of LAMP2 positive, mannose 6-phosphate receptor negative vesicles that are distinct from cholesterol enriched LAMP2 positive lysosomes, where LBPA and NPC2 reside, suggesting that transient interactions between NPC1 positive organelles and cholesterol rich lysosomes occur to effect normal cholesterol egress via NPC2-NPC1 interaction (*Neufeld et al., 1999*). While several groups have reported NPC1-independent egress of cholesterol from the LE/LY (*Boadu et al., 2012*; *Goldman and Krise, 2010*; *Kennedy et al., 2012*), it is not clear whether these function in the normal situation, or whether they are manifested secondary to NPC1 dysfunction.

Under normal conditions it is likely that the critical functional interaction between NPC2 and LBPA, demonstrated in the present studies, is followed by an interaction between NPC2 and NPC1

at the limiting lysosomal membrane, allowing for cholesterol to egress from the endolysosomal system. However, our studies have also implied that in the presence of intact NPC2, bypassing dysfunctional NPC1 may be achieved via PG-mediated or direct LBPA enrichment. LBPA enrichment may also be effective in NPC2 cases where the mutation is in a residue outside of the hydrophobic knob. For example, a human mutation in D72 has been reported to be disease causing (*Biesecker et al., 2009*), and we found here that PG supplementation/LBPA enrichment allowed NPC2 deficient cells to be effectively cleared by the D72A protein; supplementation with D72A-NPC2 was entirely ineffective prior to LBPA enrichment. In a recent high-throughput screen for drugs that raise LBPA levels, (*Moreau et al., 2019*) showed that a compound which increased LBPA levels cleared cholesterol from an NPC2 patient cell line; the mutated residue in this instance, C93, lies outside the hydrophobic knob. Thus, the in vitro evidence strongly suggests that LBPA enrichment can affectively ameliorate cellular cholesterol accumulation in the majority of NPC disease cases, including potentially all NPC1 mutations as well as NPC2 mutations outside the hydrophobic knob.

Currently there is no cure for NPC disease. Pharmacological options are limited, and palliative care remains the standard for treatment of the disease, focusing on increasing the length and quality of life for affected patients. Based on the existing in vitro evidence, we propose that cellular LBPA enrichment is worth exploring as a possible therapy. Aerosolized phospholipids such as dipalmitoyl phosphatidylcholine are widely used to enhance pulmonary drug delivery (*Duret et al., 2014*), and can be adequately nebulized without losing compositional integrity (*Schreier et al., 1994*). PG itself, administered intranasally, has been used as a therapy by the Voelker group to effectively inhibit respiratory syncytial virus infection (*Numata et al., 2010*; *Numata et al., 2013*) and influenza A virus (*Numata et al., 2012*). Moreover, lipid based colloidal carriers are able to cross the blood brain barrier (BBB) when administered intranasally (*Ganesan et al., 2018*; *Mittal et al., 2014*; *Patel and Patel, 2017*; *Tapeinos et al., 2017*) and have recently been shown to be efficient drug delivery vehicles in the treatment of intrinsic brain tumors (*van Woensel et al., 2013*) and neurodegenerative diseases including Alzheimer's (*Agrawal et al., 2018*; *Tapeinos et al., 2017*) and Parkinson's (*Tapeinos et al., 2017*; *Yang et al., 2016*). Given the historical difficulties in treating the neurological effects of NPC disease, the development of a minimally invasive, effective treatment with intrinsic ability to cross the BBB is of interest.

# Materials and methods

**Key resources table**

| Reagent type (species) or resource | Designation | Source or reference | Identifiers | Additional information |
|---|---|---|---|---|
| Cell line (*C. griseus*) | CHO-KI cells, NPC2-800#7 | PMID: 17018531 | | Dr. Peter Lobel (Robert Wood Johnson Medical School) |
| Cell line (*Homo sapiens*) | WT fibroblasts | Coriell Institute for Medical Research | Cat# GM03652; RRID:CVCL_7397 | Human skin fibroblasts from an apparently healthy 24 year old male |
| Cell line (*Homo sapiens*) | NPC2 fibroblasts | Coriell Institute for Medical Research | Cat# GM18455; RRID:CVCL_DA79 | Human skin fibroblasts from male identified as compound heterozygote at the NPC2 gene locus: results in E20X and C47F |
| Cell line (*Homo sapiens*) | NPC1 fibroblasts | Coriell Institute for Medical Research | Cat# GM03123; RRID:CVCL_7374 | Human skin fibroblasts from 9 year old female identified as compound heterozygote at the NPC1 gene locus: results in P237S and I1061T |

*Continued on next page*

*Continued*

| Reagent type (species) or resource | Designation | Source or reference | Identifiers | Additional information |
|---|---|---|---|---|
| Cell line (*Homo sapiens*) | HeLa (ATCC CCL-2) cells | ATCC | Cat # CCL-2; RRID:CVCL_0030 | Human epithelial cells from cervix of a 31 year old female with adenocarcinoma |
| Antibody | Rabbit polyclonal anti-c-myc-tag | GenScript | Cat# A00172-40; RRID:AB_914457 | IF (0.5 µg/ml) |
| Antibody | Donkey anti-rabbit IgG HRP-conjugated | GE Healthcare | Cat# NA934; RRID:AB_772206 | IF (1:20,000) |
| Antibody | Mouse monoclonal anti-myc- tag | Millipore | Cat# 05–724; RRID:AB_309938 | IF (1:2,000) |
| Antibody | Anti-mouse IgG IRDye-800CW conjugated | Li-Cor | Cat# 925–32210; RRID:AB_2687825 | IF (1:10,000) |
| Antibody | Streptavidin-d2 | Cisbio | Cat# 610SADLA | IF (50 µg/ml) |
| Antibody | Monoclonal anti-6His-Eu cryptate | Cisbio | Cat# 61HISKLA | IF (12.5 µg/ml) |
| Antibody | Rabbit monoclonal anti-NPC1 | Abcam | Cat# ab134113; RRID: AB_2734695 | WB (1:2,000) |
| Recombinant DNA reagent | Plasmid: myc 6xHis-tagged NPC2 | PMID: 12591949 | | Dr. Matthew P. Scott (Stanford University) |
| Recombinant DNA reagent | Plasmid: NPC1 CRISPR/Cas9 KO | Santa Cruz | sc-403252 | |
| Recombinant DNA reagent | Plasmid: mutant 125I-perfringolysin O (PFO*) | PMID: 23754385 | | Dr. Arun Radhakrishnan (UT Southwestern) |
| Sequence-based reagent | H31A mutant NPC2 primer: Forward, CCCACCGATCCC TGTCAGCTGGCCAAAGG; Reverse, CCTTTGGCCAGC TGAGGGATCGGTGGG | Sigma | | |
| Sequence-based reagent | D113A mutant NPC2 primer: Forward, GTGGTGGAATG GAAACTTGAAGCTGACAAAAAG; Reverse, CTTTTTGTC AGCTTCAAG TTTCCATTCCACCAC | Sigma | | |
| Sequence-based reagent | Q29A mutant NPC2 primer: Forward, CCCACCGATCCC TGTGCGCTGCACAAAGGCCAG; Reverse, CTGGCCTTT GTGCAGCGCACAG GGATCGGTGGG | Sigma | | |
| Sequence-based reagent | E108A mutant NPC2 primer: Forward, CTGGTGGTGGCA TGGAAACTTGAACTTGAAG; Reverse,CTTCAAGTTCAA GTTTCCATGCCACCACCAG | Sigma | | |
| Sequence-based reagent | D72A mutant NPC2 primer: Forward, CCCATTCCTGAG CCTGATGGTTGTAAG AGTGGAATTAAC; Reverse, GTT AATTCCACT CTTACAACCCGCAGG CTCAGGAATGGG | Sigma | | |

*Continued on next page*

*Continued*

| Reagent type (species) or resource | Designation | Source or reference | Identifiers | Additional information |
|---|---|---|---|---|
| Sequence-based reagent | H56A mutant NPC2 primer: Forward,GCCTTGGTCGC CGGCATCCTGGAAGGG; Reverse, CCCTTCCAGGAT GCCGGCGACCAAGGC | Sigma | | |
| Sequence-based reagent | G57D mutant NPC2 primer: Forward, CGGCCTTGGTCC ACGACATCCTGG; Reverse, CCAGGATGTCG TGGACCAAGGCCG | Sigma | | |
| Sequence-based reagent | I58A mutant NPC2 primer: Forward, GCCTTGGTCCAC GGCGCACTGGAAGGGATCC; Reverse, GGATCCCTTC CAGTGC GCCGTGGACCAAGGC | Sigma | | |
| Sequence-based reagent | G61A mutant NPC2 primer: Forward, GCATCCTGGAAG CGATCCGGGTCCC; Reverse, GGGACCCGGATC GCTTCCAGGATGC | Sigma | | |
| Sequence-based reagent | I62D mutant NPC2 primer: Forward, GCATCCTGGAAG GGGACCGGGTCCCCTTCC; Reverse, GGAAGGGGACC CGG TCCCCTTCCAGGATGC | Sigma | | |
| Sequence-based reagent | V64A mutant NPC2 primer: Forward, GGAAGGGATCCG GGCCCCCTTCCCTATTCC; Reverse, GGAATAGGGAAG GGGG CCCGGATCCCTTCC | Sigma | | |
| Chemical compound, drug | Cholesterol,>99% | Sigma Aldrich | Cat# C8667; CAS 57-88-5 | |
| Chemical compound, drug | Egg phosphatidylcholine (EPC) | Avanti Polar Lipids | Cat# 840051; CAS 97281-44-2 | |
| Chemical compound, drug | 18:1 Bismonoacylglycerol phosphate (BMP, aka LBPA) S,R isomer | Avanti Polar Lipids | Cat# 857133; CAS 799268-67-0 | |
| Chemical compound, drug | 18:1 Phosphatidic acid (PA) | Avanti Polar Lipids | Cat# 840875; CAS 108392-02-5 | |
| Chemical compound, drug | 18:1 Phosphatidylglycerol (PG) | Avanti Polar Lipids | Cat# 840475; CAS 67254-28-8 | |
| Chemical compound, drug | 18:1 Phosphatidylserine (PS) | Avanti Polar Lipids | Cat# 840035; CAS 90693-88-2 | |
| Chemical compound, drug | 18:1-12:0 Biotin PS | Avanti Polar Lipids | Cat# 860560; CAS 799812-66-1 | |
| Chemical compound, drug | 18:1-12:0 Biotin PA | Avanti Polar Lipids | Cat# 860561 | |
| Chemical compound, drug | 18:1-12:0 Biotin PG | Avanti Polar Lipids | Cat# 860581 | |

*Continued on next page*

*Continued*

| Reagent type (species) or resource | Designation | Source or reference | Identifiers | Additional information |
|---|---|---|---|---|
| Chemical compound, drug | 18:1-12:0 Biotin PC | Avanti Polar Lipids | Cat# 860563 | |
| Chemical compound, drug | Biotin-C12-ether LBPA | Echelon Biosciences | Cat# L-B1B12 | |
| Chemical compound, drug | Filipin III | Fisher Scientific | Cat# 62501NB; CAS 480-49-9 | Used at 0.05 mg/ml |
| Chemical compound, drug | Lipofectamine 3000 | Invitrogen | Cat# L3000- | |
| Commercial assay or kit | Stratagene QuickChange II Site Directed Mutagenesis Kit | Agilent | Cat# 200523 | |
| Commercial assay or kit | PureYield Plasmid Miniprep System | Promega | Cat# A1223 | |
| Software, algorithm | Orientation of Proteins in Membranes (OPM) | PMID: 16397007 | http://opm.phar.umich.edu/ RRID:SCR_011961 | |
| Software, algorithm | CLUSTAL Omega | PMID: 21988835 | http://www.ebi.ac.uk/ Tools/msa/clustalo/ RRID:SCR_001591 | |
| Software, algorithm | PAM250 scoring matrix | PMID: 24509512 | | |
| Software, algorithm | Kyte and Doolittle Hydropathicity scale | PMID: 7108955 | | |
| Software, algorithm | ProtScale Tool | *Gasteiger et al., 2005* | https://web.expasy.org/protscale/ | |
| Software, algorithm | ProData SX software, v2.5.0 | Applied Photophysics | https://www.photophysics.com | |
| Software, algorithm | NIS Elements BR software, v3.2 | Nikon Inc | https://www.nikoninstruments.com/Products/Software/NIS-Elements-Basic-Research RRID:SCR_014329 | |

## Orientation of NPC2 in membranes

The Orientation of Proteins in Membranes (OPM) database (http://opm.phar.umich.edu/) was used to predict spatial orientation of NPC2 protein with respect to the hydrophobic core of lipid bilayers. Protein structure of NPC2 (PDB: 1NEP) was searched against the OPM database and the resulting coordinate file and orientation predictions were obtained.

## Sequence alignment and hydrophobicity analysis

Protein sequences for human NPC2 (NCBI Accession: NP_006423.1), rat NPC2 (NCBI Accession: NP_775141.2 ) mouse NPC2 (NCBI Accession: NP_075898.1), bovine NPC2 (NCBI Accession: NP_776343.1), cat NPC2 (NCBI Accession: XP_003987882.1), chimpanzee NPC2 (NCBI Accession: NP_001009075.1) and the yeast NPC2 (NCBI Accession: KZV12184.1) were aligned with CLUSTAL Omega (*Sievers et al., 2011*). Protein conservation was scored using a PAM250 scoring matrix, which is extrapolated from comparisons of closely related proteins, similar to the current application (*Pearson, 2013*). Domain specific conservation of the hydrophobic knob between each NPC2 sequence was analyzed by taking the sum of the conservation scores of each residue from 56 to 64, relative to the human NPC2 sequence. Protein hydrophobicity was scored using the Kyte and Doolittle Amino acid Hydropathicity scale (*Kyte and Doolittle, 1982*). Domain specific hydrophobicity of the hydrophobic knob of each NPC2 sequence was analyzed by taking the sum of the Kyte and Doolittle Amino acid Hydropathicity score of each residue from 56 to 64. Whole protein hydrophobicity of aligned sequences were analyzed using the ProtScale Tool on the ExPASy server (*Gasteiger et al., 2005*), based on the Kyte and Doolittle Amino acid Hydropathicity scale (*Kyte and Doolittle, 1982*).

## Cell lines

Chinese hamster ovary (CHO) cells transfected with a human NPC2 expression vector (NPC2-800#7) (*Liou et al., 2006*), kindly provided by Peter Lobel, were maintained in F12-K media (Invitrogen) supplemented with 10% FBS and 1 mg/mL gentamycin. Human WT (GM03652), NPC1 (GM03123), and NPC2 (GM18455) fibroblasts (Coriell Institute, Camden, NJ) and HeLa (ATTC CCL-2) (ATCC, Manassas, VA) cells were maintained in DMEM media (Invitrogen) supplemented with 15% FBS and 1% penicillin-streptomycin. All cells were at passage 18 or below. Authentication of all fibroblasts and HeLa CCL-2 cells was obtained via STR analysis. All cultures were confirmed mycoplasma free at receipt and were cultured aseptically using only mycoplasma free reagents.

## Generation of HeLa NPC1 knockout

NPC1 CRISPR-Cas9 KO construct was purchased from Santa Cruz Biotechnology (Dallas, TX) and transfected into HeLa CCL-2 cells using the Lipofectamine 3000 reagent (Invitrogen). Cells were selected with puromycin at 2 µg/ml for 4 days. Single cell clones were isolated and subsequently screened for loss of NPC1 expression using Western Blot analysis with an anti-NPC1 antibody (Abcam, Cambridge, MA).

## Generation and isolation of NPC2 mutants

Point mutations were created with the Stratagene QuikChange Site Directed Mutagenesis Kit (Agilent, Santa Clara, CA), using myc 6xHis-tagged murine NPC2 plasmid, according to the manufacturer's directions and as described previously (*Ko et al., 2003*; *McCauliff et al., 2015*). Plasmid isolation was performed using the PureYield Plasmid Miniprep system (Promega, Madison, WI). Wild type and mutant myc 6xHis-tagged NPC2 proteins were purified from transfected NPC2-800#7 CHO cells, which secrete large amounts of the NPC2 protein into CD CHO media (Invitrogen), using a 10 kDa cutoff flow filtration membrane (Millipore, Bedford, MA) to initially concentrate the media, as previously described (*McCauliff et al., 2015*). The presence of purified NPC2 was confirmed by SDS-PAGE (*Cheruku et al., 2006*; *Ko et al., 2003*; *McCauliff et al., 2015*); proteins used were $\geq$90% pure by silver staining. Buffer exchange was performed using Sartorius Vivaspin Turbo four filters with a 10 kDa cutoff membrane followed by dilution in sodium citrate buffer (in 20 mM sodium citrate, 150 mM NaCl, pH 5.0).

## Cholesterol binding by NPC2

Equilibrium binding constants for cholesterol binding by WT and mutant NPC2 proteins were determined by quenching of tryptophan emission, as previously described (*Cheruku et al., 2006*; *Ko et al., 2003*; *McCauliff et al., 2015*; *Xu et al., 2008*). Purified proteins were delipidated via acetone precipitation (*Liou et al., 2006*) and resuspended in sodium citrate buffer. The delipidated NPC2 proteins were incubated with increasing concentrations of cholesterol (>99%) (Sigma Aldrich), in DMSO, for 20 min at 25°C and tryptophan emission spectrum were acquired on an SLM fluorimeter (Horiba Jobin Yvon, Edison, NJ). Final DMSO concentration was >1% (v/v). AUC were determined for all spectrum and binding constants were determined by hyperbolic fit of the data using Sigma Plot software (San Jose, CA).

## Membrane vesicle preparation

Small unilamellar vesicles (SUV) were prepared by sonication and ultracentrifugation as previously described (*Storch and Kleinfeld, 1986*). Large unilamellar vesicles (LUV) were prepared via freeze-thaw cycling and extrusion through a 100 nm membrane, as previously described (*McCauliff et al., 2015*; *Xu et al., 2008*). The final phospholipid concentration of all vesicles was determined by quantification of inorganic phosphate (*Gomori, 1942*). Vesicles were maintained above the phase transition temperatures of all constituent lipids. Standard vesicles were composed of 100 mol% egg phosphatidyl choline (EPC) (Avanti Polar Lipids, Alabaster, AL). Where noted, LBPA (Avanti) replaced 5–25 mol% of EPC in SUV and/or LUV preparations, as indicated. All vesicles used for in vitro transfer assays were prepared in sodium citrate buffer, pH 5.0. For incubation with cells, vesicles were composed of 100 mol% PG (Avanti), 25 mol% LBPA or 100 mol% PC (Avanti) and prepared in sterile phosphate-buffered saline, pH 7.4.

## Effect of LBPA on cholesterol transfer by NPC2

As detailed previously (*McCauliff et al., 2011*; *McCauliff et al., 2015*; *Xu et al., 2008*), transfer of cholesterol from WT or mutant NPC2 protein to membranes was monitored by the dequenching of tryptophan fluorescence over time using a stopped-flow mixing chamber interfaced with a Spectro-fluormeter SX20 (Applied Photophysics, Leatherhead, UK). Cholesterol transfer rates from 1 µM WT NPC2 to 125 µM EPC membranes containing increasing mol percentages of LBPA (0%, 10%, 20% and 30%) were determined at 25°C. Additionally, to determine the effects of LBPA on the sterol transport properties of mutant NPC2 proteins, transfer of cholesterol from 1 µM WT or mutant NPC2 to 125 µM SUVs composed of either 100% EPC or 25 mol% LBPA/EPC was monitored at 25°C. Instrument settings to ensure the absence of photobleaching were established before each experiment. Data were analyzed with the Pro-Data SX software provided with the Applied Photophysics stopped-flow spectrofluorometer, and the cholesterol transfer rates were obtained by single exponential fitting of the curves, as previously described (*McCauliff et al., 2015*; *Xu et al., 2008*).

## Equilibrium distribution of $^3$H-cholesterol secondary to transfer between NPC2 and membrane vesicles

Large unilamellar vesicles (LUVs) were prepared by extrusion as described previously (*Wootan and Storch, 1994*). Vesicles were composed of 100% EPC or 75% EPC/25% LBPA (mol/mol), with a trace amount of $^{14}$C-cholesterol ester as a nonexchangeable marker of the LUVs. $^3$H-cholesterol was incubated with NPC2 for $\geq$15 min, and the complex mixed with LUVs. The LUV were pelleted after 30 s or 5 min by ultracentrifugation at 100,000 x g for 45 min. For the reverse reaction, the LUVs contained $^3$H-cholesterol and were mixed with apo NPC2. Following correction for unpelleted LUVs and for NPC2 in the pellet, determined by tryptophan emission, the relative distribution of $^3$H-cholesterol between NPC2 and phospholipid membranes was determined (*Storch and Bass, 1990*; *Xu et al., 2008*).

## Membrane-membrane interaction

Effects of NPC2 on vesicle-vesicle interactions were assessed in two ways, both using light scattering approaches. 200 µM LUVs were mixed with 1 µM WT or mutant NPC2 proteins in a 96-well plate reader, and absorbance at 350 nm monitored every 10 s over a period of 30 min (*Petruševska et al., 2013*). Increases in A350nm (light scattering) are indicative of vesicle-vesicle interaction, the rate of which was determined by a three-parameter hyperbolic fit of the data using Sigma Plot software. Additionally, 750 µM LUVs were mixed with 1 µM WT or mutant NPC2 proteins in a spectrophotometer (Hitachi U-2900, Pleasanton, CA) and A350nm was measured over a period of 60 s (*Schulz et al., 2009*); rates of vesicle-vesicle interaction were obtained by single exponential fitting of the curves.

## Lipid blot analysis of NPC2 interactions

To assess WT NPC2 binding to various LBPA isomers, LBPA Snoopers (Avanti), containing 1 µg spots of pure LBPA isomers, were blocked with tris-buffered saline (TBS) (0.8% NaCl, 20 mM Tris-HCl pH 7.4) + 3% BSA (fatty-acid free), followed by a one hour incubation at room temperature with 5 µg of WT NPC2 in TBS pH 7.4 + 3% BSA, at a final concentration of 0.5 µg/ml protein. The protein solution was removed and the Snoopers were washed with TBS. NPC2 bound to LBPA isomers was detected by incubating the Snoopers with rabbit polyclonal anti-c-myc-tag antibody (GenScript, Piscataway, NJ) at a concentration of 0.5 µg/ml in TBS + 3% BSA for one hour at room temperature. Following removal of the primary antibody, the strips were washed with TBS and incubated with anti-rabbit IgG HRP-conjugated antibodies (GE Healthcare,Pittsburg, PA) at a 1:20,000 dilution in TBS + 3% BSA. After a one-hour incubation with the secondary antibody, the Snoopers were washed with TBS + 0.05% Tween and developed with ECL reagents (GE Healthcare).

For further analysis of NPC2-lipid interaction, Hybond-C membranes (GE-Healthcare) were spotted with either 500 pmol of 18:1 LBPA/BMP (S,R) (Avanti), 18:1 PA (Avanti), 18:1 PG (Avanti), 18:1 PS (Avanti), and Egg PC, or with increasing concentrations of LBPA (125, 250, 375 and 500 pmol) to analyze binding of WT NPC2 to various phospholipid species, or binding of NPC2 mutants to LBPA, respectively. Following the protocol of *Dowler et al. (2002)*, each phospholipid was spotted in duplicate and allowed to dry for one hour. Membranes were blocked for 1 hr in blocking buffer

containing TBS (50 mM Tris/HCl, pH 7.5, 150 mM NaCl) and 5% (w/v) non-fat dry milk. Membranes were then incubated overnight at 4°C with either WT or mutant NPC2 protein diluted to a final concentration of 1 ug/ml in TBS and 3% (w/v) non-fat dry milk. The membranes were then washed at room temperature in TBST (0.1% Tween 20) six times for 5 min each, followed by incubation with mouse monoclonal anti-myc antibody (Millipore) at a 1:2000 dilution in TBS and 3% (w/v) milk. After washing with TBST, the membrane was then incubated with anti-mouse IgG IRDye-800CW conjugated antibody (LI-COR, Lincoln, NE) at a 1:10,000 dilution in TBS, 0.1% SDS, and 3% (w/v) milk. The membranes were finally washed in TBST 12 times for 5 min each at room temperature before acquiring images on the LI-COR Odyssey.

## Homogenous Time Resolved Fluorescence (HTRF)

Assays were performed as described in *Fleury et al. (2015)*, with minor modifications. Reaction mixtures for the interaction assays were prepared in white 384-well polystyrene non-binding surface NBS microplates (Corning, Corning, NY) with a final volume of 20 µL per well. Each reaction mix contained 6 µL of buffer A (20 mM sodium citrate, 150 mM NaCl, 1 mM EDTA pH 5.0), 2 µL of the recombinant WT or mutant His-tagged NPC2 at a final concentration of 75 nM, 2 µL of biotinylated lipid solution in buffer A at a final concentration range of 1 µM - 58.5 nM, 5 µL of streptavidin-d2 conjugate (Cisbio Bioassays, Bedford, MA) and 5 µL of monoclonal anti-6His-europium cryptate antibody (Cisbio) in detection buffer (20 mM of HEPES pH 8.5, 200 mM of potassium fluoride, 1% bovine serum albumin). The biotinylated lipids (PS, PA, PG, PC and LPBA; Avanti) were dried under nitrogen and the resultant film was initially reconstituted in EtOH and secondarily in binding buffer at a ratio of 1:10 EtOH:buffer. The final concentration of ethanol in the reaction was 1% (v/v). Following an 18 hr incubation of the reaction mixture at room temperature, the fluorescence was measured with an Envision plate reader (Perkin Elmer; λex = 320 nm, λem = 615 and 665 nm; 100 µs delay time). The HTRF ratio value was represented as Ch1/Ch2*10,000 where Ch1 is the energy transfer signal at 665 nm, and Ch2 is the europium cryptate antibody signal at 615 nm. The negative control wells contained donor and acceptor fluorochromes without NPC2 or biotinylated lipid. The negative control (background) readout was subtracted from all the sample readings.

## Clearance of cellular cholesterol by NPC2 proteins

As detailed previously, a single dose of purified WT or mutant NPC2 protein was added to the media of NPC2 mutant fibroblasts cultured on 8-well tissue culture slides (Nalgene), and allowed to incubate for 3 days. The final concentration of added protein was 0.4 nM. In keeping with prior literature (*Ko et al., 2003*; *Liou et al., 2006*; *McCauliff et al., 2015*; *Wang et al., 2010*), since identical amounts of NPC2 protein were added to equivalent samples of cultured cells, we reasonably assume equivalent uptake of the various NPC2 proteins; notably all bind cholesterol similarly, implying that they fold normally and thus are of similar shape and size, precluding any potential difference in uptake via fluid phase endocytosis. Following incubations the cells were fixed and stained with 0.05 mg/mL filipin III (Fisher) and subsequently imaged on a Nikon Eclipse E800 epifluorescence microscope using a DAPI filter set. Filipin stain was quantified as a ratio of fluorescence intensity per unit cell area in treated and untreated conditions with the accompanying NIS-Elements software (Nikon Inc). Results are corrected for background fluorescence and are representative of an average of 80 to 100 cells per condition.

## Cellular LBPA enrichment via PG supplementation

WT, NPC1, and NPC2 mutant fibroblasts were cultured to confluence in 100 mm petri dishes and passaged by trypsinization at a 1:3 ratio. After 24–48 hr, media was removed and replaced by media supplemented with either 30, 100, or 250 µM PG SUVs (*Bouvier et al., 2009*; *Luquain-Costaz et al., 2013*), or with 100 µM LBPA or PC SUVs. After 24 hr, cells were collected and 4–6 dishes of the same treatment were pooled. Protein levels were analyzed using the Bradford method (*Bradford, 1976*). Total cell lipids were extracted from 2 mL of 1 mg/mL protein via the method of *Bligh and Dyer (1959)*, resuspended in 200 µL 2:1 chloroform:methanol, and were run on HPTLC plates (EMD Chemicals, Inc) in a solvent of 65:35:5 chloroform:methanol(v/v):30% ammonium hydroxide(v/v) (*Akgoc et al., 2015*). Lipid spots were quantified by densiometric analysis (ImageJ) from standard curves of authentic standards.

## Clearance of cholesterol by PG supplementation

WT, NPC1–, and NPC2 mutant fibroblasts were plated onto 8-well tissue culture slides (BD falcon) at a density of approximately 20,000 cells per well and incubated at 37°C, 5% $CO_2$ for 24 hr. Culture media was then removed and the cells were incubated with media supplemented with 100 µM PG SUVs for 24 hr. Cells were subsequently fixed and stained with 0.05 mg/mL filipin III. In some experiments, NPC2 deficient fibroblasts were secondarily incubated for 24 hr with WT or mutant NPC2 proteins at a final concentration of 0.4 nM, 24 hr after the 24 hr supplementation with PG. Cells were imaged on a Nikon Eclipse E800 epifluorescence microscope using a DAPI filter set to detect filipin. Filipin and antibody stain intensity was quantified with the accompanying NIS-Elements software (Nikon Inc); cholesterol accumulation was calculated as the ratio of filipin stain intensity to cell area (*Delton-Vandenbroucke et al., 2007*; *McCauliff et al., 2011*; *McCauliff et al., 2015*).

HeLa NPC1 knockout cells were seeded in 6 cm dishes at a density of $2 \times 10^5$ per dish, in DMEM (Sigma) supplemented with 10% FBS, 1% Penicillin-Streptomycin. Forty-eight hours after seeding, cells were incubated with 100 µM PG SUVs or vehicle (PBS) for 24 to 48 hr. One dish of cells per treatment condition was analyzed for cholesterol accumulation via mutant 125I-perfringolysin O (PFO*) staining. PFO* plasmid was kindly provided by Arun Radhakrishnan (University of Texas Southwestern, Dallas) (*Das et al., 2013*) and PFO* purification and staining was performed as described by *Li et al. (2017)*. Stained cells in PBS were analyzed using a BD Acuri TM C6 flow cytometer.

WT fibroblasts cells were seeded in 6 cm dishes at a density of $1.2 \times 10^5$ cells per dish, in EMEM (Sigma) containing Earle's Salts and Nonessential Amino Acids supplemented with 15% FBS, 1% Penicillin-Streptomycin, 2 mM L-glutamine and 1 mM sodium pyruvate. Forty-eight hours after seeding, cells were incubated with 1 µM U18666A in DMSO (0.1% v/v) for 24 hr in order to induce an NPC1-like phenotype. Cells were then incubated with 100 µM PG SUVs for 16, 34 and 40 hr without changing media. Following incubation with PG SUVs, two dishes of cells per sample were trypsinized, pelleted and used for PFO* labeling for flow cytometry as described above.

## Statistical analysis

Statistical analysis was performed using OriginPro 2016 (OriginLab Corporation) and SigmaPlot 12.0. Means were compared using Student's *t*-test for independent samples or one-way ANOVA where indicated with $p < 0.05$ considered as significantly different.

## Acknowledgements

This work was supported by funds from the Ara Parseghian Medical Research Foundation (JS, OI), the American Heart Association (predoctoral fellowship 11PRE7330012 to LAM, career development grant 18CDA34110230 to OI, and Grant-in-Aid 14GRNT19990014 to JS), and National Institutes of Health GM 1125866 (JS). The authors would like to thank Joseph Nickels and Hsing-Yin Liu for their assistance with the HTRF assay, and Peter Lobel for generously providing the NPC2-800#7 CHO cells.

## Additional information

### Funding

| Funder | Grant reference number | Author |
|---|---|---|
| Ara Parseghian Medical Research Foundation | | Olga Ilnytska<br>Judith Storch |
| American Heart Association | 11PRE7330012 | Leslie A McCauliff |
| American Heart Association | 18CDA34110230 | Olga Ilnytska |
| American Heart Association | 14GRNT19990014 | Judith Storch |
| National Institutes of Health | GM 1125866 | Judith Storch |

The funders had no role in study design, data collection and interpretation, or the decision to submit the work for publication.

## Author contributions

Leslie A McCauliff, Conceptualization, Formal analysis, Funding acquisition, Investigation, Visualization, Methodology, Writing—original draft, Writing—review and editing; Annette Langan, Formal analysis, Investigation, Visualization, Writing—original draft, Writing—review and editing; Ran Li, Formal analysis, Investigation, Visualization, Methodology, Writing—original draft, Writing—review and editing; Olga Ilnytska, Investigation, Visualization, Methodology, Writing—review and editing, Funding acquisition; Debosreeta Bose, Miriam Waghalter, Kimberly Lai, Investigation, Writing—review and editing; Peter C Kahn, Supervision, Writing—review and editing; Judith Storch, Conceptualization, Resources, Formal analysis, Supervision, Funding acquisition, Validation, Visualization, Writing—original draft, Project administration, Writing—review and editing

## Author ORCIDs

Leslie A McCauliff https://orcid.org/0000-0002-5744-2737
Judith Storch https://orcid.org/0000-0001-5482-1777

## Decision letter and Author response

Decision letter https://doi.org/10.7554/eLife.50832.029
Author response https://doi.org/10.7554/eLife.50832.030

# Additional files

## Supplementary files

- Transparent reporting form DOI: https://doi.org/10.7554/eLife.50832.013

## Data availability

All data generated or analysed during this study are included in the manuscript and supporting files.

The following previously published datasets were used:

| Author(s) | Year | Dataset title | Dataset URL | Database and Identifier |
|---|---|---|---|---|
| Lomize MA, Lomize AL, Pogozheva ID, Mosberg HI | 2006 | 1nep, Epididymal secretory protein E1 | https://opm.phar.umich.edu/proteins/271 | Orientation of Proteins in Membranes (OPM) Database, 1nep |
| National Library of Medicine (US), National Center for Biotechnology Information | 1988 | NPC intracellular cholesterol transporter 2 isoform 2 precursor [Homo sapiens] | https://www.ncbi.nlm.nih.gov/protein/NP_006423.1 | NCBI Protein Database, NP_006423.1 |
| National Library of Medicine (US), National Center for Biotechnology Information | 1988 | NPC intracellular cholesterol transporter 2 precursor [Mus musculus] | https://www.ncbi.nlm.nih.gov/protein/NP_075898.1 | NCBI Protein Database, NP_075898.1 |
| National Library of Medicine (US), National Center for Biotechnology Information | 1988 | NPC intracellular cholesterol transporter 2 precursor [Bos taurus] | https://www.ncbi.nlm.nih.gov/protein/NP_776343.1 | NCBI Protein Database, NP_776343.1 |
| National Library of Medicine (US), National Center for Biotechnology Information | 1988 | epididymal secretory protein E1 [Felis catus] | https://www.ncbi.nlm.nih.gov/protein/XP_003987882.1 | NCBI Protein Database, XP_003987882.1 |
| National Library of Medicine (US), National Center for Biotechnology Information | 1988 | NPC intracellular cholesterol transporter 2 precursor [Pan troglodytes] | https://www.ncbi.nlm.nih.gov/protein/NP_001009075.1 | NCBI Protein Database, NP_001009075.1 |

| National Library of Medicine (US), National Center for Biotechnology Information | 1988 | NPC2 [*Saccharomyces cerevisiae*] | https://www.ncbi.nlm.nih.gov/protein/KZV12184.1 | NCBI Protein Database, KZV12184.1 |

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
