## [Decision Letter]

Thank you for submitting your work entitled "Interaction of NPC2 protein with Lysobisphosphatidic Acid is required for normal endolysosomal cholesterol trafficking" for consideration by *eLife*. Your article has been reviewed by three peer reviewers, including Suzanne Pfeffer as the Reviewing Editor and Reviewer #1, and the evaluation has been overseen by a Senior Editor.

This decision has been reached after consultation between the reviewers. Based on these discussions and the individual reviews below, we regret to inform you that your work will not be considered further for publication in *eLife*. We realize that this news will be disappointing. The reviewers share a strong interest in understanding how cholesterol is exported from lysosomes, and all felt that it would be spectacular if you were able to show how LBPA, in concert with NPC2, drives NPC1-independent cholesterol export from lysosomes. Unfortunately, the present experiments are not yet at that stage, and were altogether not deemed novel enough to warrant presentation in *eLife* at this time.

Reviewer #1:

This paper shows for the first time that NPC2 binds directly to LBPA via a hydrophobic knob, using solid phase filter assays and a solution based FRET read-out of proximity of a biotinylated lipid to an antibody bound, his-tagged NPC2 protein. The authors show that the ability of NPC2 to bind LBPA via these residues correlated with cholesterol export in cells. This work is for the most part carried out to a high standard and will be of interest to those studying cholesterol regulation and trafficking after consideration of the following points.

1) The authors use a 3 day, added NPC2 protein-rescue of cholesterol accumulation monitoring filipin area rather than intensity. Do the area and intensity metrics agree? This is important, as it seems that intensity would be more reflective of actual amounts rather than area.

2) Does NPC2 float with liposomes containing LBPA? Apologies if this was in a previous paper, but it would help the reader to know this as it adds credence to the direct binding argument.

3) The authors discuss NPC1-independent routes of cholesterol export. It might help the reader to know how deep in the endocytic pathway LBPA, NPC2 and NPC1 are each found, which could account for some of the differences in their activities. Also, MVB exocytosis may be LBPA-stimulated and could be discussed.

Figure 5C needs color as it is impossible to decipher

Reviewer #2:

The authors have studied NPC2 and LBPA for many years and have published several papers on this topic. In this study, they report on the existence of an NPC2-dependent but NPC1-independent pathway of cholesterol egress from lysosomes. They do not claim that the extensively-characterized NPC2-NPC1 hydrophobic handoff model does not exist, but rather that there is also an NPC1-independent pathway. This alternative pathway is crucially dependent on the lipid, LBPA, which is enriched in lysosomal membranes. The authors report mutations in NPC2 near a putative hydrophobic knob that are responsible for this LBPA-dependent transport pathway.

The new findings here rest on the characterization of mutations in the hydrophobic knob region that are important in the LBPA-dependent transfer of cholesterol from NPC2 to membranes in a NPC1-independent manner. However, the experimental results do not fully support this, and the following points need addressing.

1) Transfer of cholesterol from NPC2 to liposomes is measured by dequenching the tryptophan fluorescence of NPC2 (which is quenched by the bound cholesterol). This does not show that cholesterol has been transferred to the liposome. While this is the likely outcome, direct measurement of transfer of cholesterol needs to be done (with radiolabeled cholesterol or analysis of cholesterol in liposomes by other means). This is a critical point of this paper.

2) The central point of this paper is that the LBPA-mediated pathway is independent of NPC1. The authors need to show the hydrophobic knob in the context of the NPC2/NPC1 hydrophobic tunnel (Li et al., 2016), and need to show the effect of their LBPA-sensitive mutations on NPC1-mediated cholesterol transfer, using the methodology of Wang et al., 2010. Without this, it is premature to say that the LBPA-dependent pathway is NPC1-independent.

3) Does binding to LBPA affect the binding to cholesterol? According to the author's model, one might expect that LBPA might displace the bound cholesterol from NPC2 so that it can be transferred to liposomes.

Reviewer #3:

This is a well-documented paper on the interaction of NPC2 protein with LBPA (otherwise known as BMP). However, given the previous publications from this group and others, I don't find the results particularly novel. It is more like working out some additional details from further mutations. Most importantly, I do not get from this paper a clear understanding of how NPC2 binding to LBPA actually leads to enhanced efflux of cholesterol from LE/LY with or without NPC1.

Essential revisions:

1) The mouse NPC2 sequence in Figure 1 shows a glutamate in the middle of the hydrophobic knob sequence. Is the mouse protein defective in transport? If not, why?

2) In measuring the amounts of filipin labeling (Figure 6), they should measure the filipin intensity rather than the area of filipin labeling. This would be a much better indicator of the amount of cholesterol in the membranes.

3) The same is true for measuring LBPA (Figure 7). Intensity measurements are made easily with any image analysis software package.

4) Much of the Discussion section seemed repetitive of earlier sections. More importantly, it did not provide mechanisms that I understood for enhanced cholesterol efflux when LBPA and/or NPC2 levels were increased.

A) How would membrane:membrane interactions of LBPA-enriched membranes facilitate efflux? Most papers have suggested that LBPA is mainly on internal membranes and not on the limiting membrane. Contacts with the limiting membrane would need to clear areas of the glycocalyx, which is considered to protect the limiting membrane against hydrolysis. Merely being near the limiting membrane but bound to a LAMP protein would still leave a large aqueous gap to be traversed by cholesterol.

[Editors’ note: what now follows is the decision letter after the authors submitted for further consideration.]

Thank you for resubmitting your work entitled "Intracellular cholesterol trafficking is dependent upon NPC2 interaction with Lysobisphosphatidic Acid" for further consideration at *eLife*. Your revised article has been evaluated by Suzanne Pfeffer (Senior and Reviewing Editor), and two other reviewers (including Fred Maxfield). You will be pleased to learn that the overall review was positive and that at this stage mostly textual changes are requested.

This manuscript reports an interesting and important finding, namely that NPC2 requires LBPA (and vice versa) to mediate cholesterol export from lysosomes. The significance of the findings is more clearly elaborated in this version. The story should be published in *eLife* after consideration of the following points.

Essential revisions:

1) Please show quantitation of at least some of the key NPC2 mutants (vs. wild type) in terms of their actual endocytosed levels and correlated abilities to correct the filipin staining phenotype shown in Figure 9.

2) Please discuss more fully the NPC1-dependence or -independence of the LBPA/NPC2 pathway. A lingering concern is the relevance of this NPC2/LBPA pathway to cholesterol trafficking. In response to reviewer 3's point 5, the authors say that "the contribution of this pathway may be either very small or possibly non-existent under "normal conditions, but in NPC1 disease the accumulation of LBPA may be a compensatory mechanism...". Yet, the Title says "Intracellular cholesterol trafficking is dependent upon NPC2 interaction with LBPA". Please discuss more clearly in the Discussion section. The authors say that studying the effects of the LBPA-sensitive NPC2 mutants on NPC1-mediated cholesterol transfer is tangential to their main point that LBPA acts through NPC2. However, their conclusions would be strengthened by studying these mutants to address NPC1-independence or dependence (OPTIONAL). "Potential NPC2 binding site on NPC1"? Li et al., 2016 should also be cited in the Introduction; Deffieu et al., 2011 showed direct binding of NPC2 to NPC1 and Li et al., 2016 measured Kds for this interaction and defined corresponding residues needed for the interaction. The binding site is pretty clear from the crystal structures and mutagenesis, or?

---

## [Author Response]

[…] The reviewers share a strong interest in understanding how cholesterol is exported from lysosomes, and all felt that it would be spectacular if you were able to show how LBPA, in concert with NPC2, drives NPC1-independent cholesterol export from lysosomes. Unfortunately, the present experiments are not yet at that stage, and were altogether not deemed novel enough to warrant presentation in eLife at this time.

We very much appreciate the helpful and encouraging review of our manuscript and the generous opportunity to resubmit an amended version in which we have attempted to respond to the suggestions and questions. We have added additional experimental evidence and, moreover, have trimmed and modified both the introduction and discussion sections as well as altered the abstract and title. We hope we have now done a better job of explaining what we consider to be a novel aspect of intracellular cholesterol trafficking within the LE/LY compartment. Point by point responses to each comment are as follows:

Reviewer #1:

This paper shows for the first time that NPC2 binds directly to LBPA via a hydrophobic knob, using solid phase filter assays and a solution based FRET read-out of proximity of a biotinylated lipid to an antibody bound, his-tagged NPC2 protein. The authors show that the ability of NPC2 to bind LBPA via these residues correlated with cholesterol export in cells. This work is for the most part carried out to a high standard and will be of interest to those studying cholesterol regulation and trafficking after consideration of the following points.1) The authors use a 3 day, added NPC2 protein-rescue of cholesterol accumulation monitoring filipin area rather than intensity. Do the area and intensity metrics agree? This is important, as it seems that intensity would be more reflective of actual amounts rather than area.

Cholesterol accumulation in treated and untreated cells was in fact measured by the intensity of filipin stain per unit area, and we have corrected the text describing our methodology, as well as appropriate figure legends, to indicate that. In previous studies, we did evaluate cholesterol accumulation as a ratio of filipin stain area to total cell area (for example McCauliff et al., 2015). As a point of information, we have consistently found that the intensity measurements, as performed in the present studies, agree with area measurements in samples analyzed both ways.

2) Does NPC2 float with liposomes containing LBPA? Apologies if this was in a previous paper, but it would help the reader to know this as it adds credence to the direct binding argument.

The question as to whether NPC2 floats with liposomes containing LBPA is an interesting and important point. As the reviewer surmised, we had in fact addressed this question in a previous publication (Xu et al., 2008), showing that the tryptophan emission of NPC2 was quenched in the presence of liposomes, indicative of protein-membrane interaction, and the apparent Kd for binding of liposomes by NPC2 obtained using these methods was reported. We agree that these previous data lend support to the present studies and have thus included a brief description of the results in subsection “Predicted orientation of NPC2 in membranes”.

3) The authors discuss NPC1-independent routes of cholesterol export. It might help the reader to know how deep in the endocytic pathway LBPA, NPC2 and NPC1 are each found, which could account for some of the differences in their activities. Also, MVB exocytosis may be LBPA-stimulated and could be discussed.

We appreciate the thoughtful suggestion of addressing NPC1, NPC2 and LBPA localization in the endocytic pathway. We agree that this point, in addition to addressing LBPA’s potential role in MVB exocytosis, would provide a greater understanding of how the present work sheds light on potential mechanisms of LE/LY cholesterol export. In the discussion of pathways of LE/LY cholesterol egress, in the Discussion section, we have now incorporated comments on the localization and overlap of LBPA and the NPC proteins in the endo/lysosomal system. Also, in the Discussion section we have addressed studies not only indicating LBPA involvement in exosome formation but also the identification of both LBPA and cholesterol in these bodies (where we address other roles of LBPA in the endo/lysosomal system).

Figure 5C needs color as it is impossible to decipher

At the reviewer’s useful suggestion, we have included color in Figure 5C for enhanced readability.

Reviewer #2:

[…] The new findings here rest on the characterization of mutations in the hydrophobic knob region that are important in the LBPA-dependent transfer of cholesterol from NPC2 to membranes in a NPC1-independent manner. However, the experimental results do not fully support this, and the following points need addressing.1) Transfer of cholesterol from NPC2 to liposomes is measured by dequenching the tryptophan fluorescence of NPC2 (which is quenched by the bound cholesterol). This does not show that cholesterol has been transferred to the liposome. While this is the likely outcome, direct measurement of transfer of cholesterol needs to be done (with radiolabeled cholesterol or analysis of cholesterol in liposomes by other means). This is a critical point of this paper.

We are confident that the observed changes in NPC2 tryptophan fluorescence are indicative of cholesterol transfer between the protein and liposomes: In our studies of cholesterol movement between NPC2 and membranes, we have shown that the directionality of the transfer is reflected by opposing behavior of the tryptophan signal, where de-quenching is observed when NPC2 acts as the cholesterol donor versus quenching when NPC2 is the sterol acceptor, as demonstrated in Cheruku et al., (2006) and Xu et al., (2008). Such opposing fluorescence intensity changes would not occur if transfer of sterol on or off of NPC2 were not occurring; if sterol were not transferring and the protein and membranes were interacting, the fluorescence changes would be similar whether the cholesterol was initially bound to the protein or to the membranes.

It is also worth noting that such fluorescence-based methodologies have been used for decades to monitor protein-ligand binding and transfer kinetics (Atkinson et al., 2004; Bian and De Camilli, 2019; Nemecz et al., 1988; Nichols and Pagano, 1983); arguably this approach is preferable to traditional equilibrium distribution assays in which physical separation of donor and acceptor populations is relatively slow and cannot be used to determine the kinetics of very rapid transfer, as is the case in the present studies.

To support the interpretation that the fluorescence dequenching is indicating cholesterol transfer from NPC2 to membranes, we have conducted in vitro assays using ^3^H-cholesterol to monitor transfer from holo-NPC2 to vesicles, and from ^3^H-cholesterol-containing membranes to apo-NPC2. In these experiments NPC2 and membranes were separated by ultracentrifugation, as described in the Materials and methods section. The results show that the labeled cholesterol transfers from the protein to the membranes and the reverse, from membranes to protein. We have included the partition coefficient which describes the distribution of sterol at equilibrium in subsection “LBPA markedly stimulates sterol transfer rates between NPC2 and membranes”

2) The central point of this paper is that the LBPA-mediated pathway is independent of NPC1. The authors need to show the hydrophobic knob in the context of the NPC2/NPC1 hydrophobic tunnel (Li et al., 2016), and need to show the effect of their LBPA-sensitive mutations on NPC1-mediated cholesterol transfer, using the methodology of Wang et al., 2010. Without this, it is premature to say that the LBPA-dependent pathway is NPC1-independent.

The reviewer raises a valid issue regarding the ability to declare that the pathway utilized by NPC2 and LBPA is independent of NPC1, as our cellular model (Coriell GM03123; compound heterozygote, C709T and T3182C) does indeed express the NPC1 protein, though defective in its ability to traffic cholesterol normally. Another interesting point is the fact that there is overlap between the LBPA-interactive domain identified here and the NPC1interactive domain on NPC2 identified by Wang et al., (2010). To further support the apparently NPC1-independent egress of cholesterol by LBPA enrichment shown here and in Chevallier et al., (2008), we have added data from two new experiments, now included as Figures 8C and D. First, we generated HeLa cells where NPC1 expression was eliminated using CRISPR-Cas9, and show that in in these cells, incubation with PG leads to significantly reduced cholesterol levels. Secondly, we used the U18666A compound, known to generate a phenocopy of NPC disease at the cellular level and recently demonstrated to target NPC1 specifically (Lu et al., 2015); here too we find that PG incubation leads to significant reductions in cellular cholesterol levels. The aforementioned limitation of our studies using the NPC1 line from Coriell has been mentioned in subsection “PG supplementation/LBPA enrichment reduces cholesterol accumulation in NPC1- but not NPC2-deficient 289 fibroblasts” in the revised manuscript. Two additional researchers, who aided in conducting these studies, have also been added to the authors’ list and their contributions cited appropriately.

We respectfully wish to point out that the highly novel aspect of our findings is that there is an obligate interaction between NPC2 and LBPA that is required for cholesterol egress from the LE/LY. Our main point is not really the existence of a potentially NPC1independent mechanism of cholesterol egress, and indeed we are not the first to propose this possibility; three different groups that have also shown the apparently NPC1-independent egress of cholesterol are cited on page 16 of the manuscript. In the case of LBPA, it was known 10 years ago that increasing the LBPA content of NPC1 deficient cells could clear the accumulated cholesterol (Chevallier et al., 2008), but the mechanism of this ‘rescue’ was not known. In this manuscript we present the discovery that the mechanism of LBPA action is critically dependent on functional NPC2 protein. This is the case whether the cholesterol egress stimulation is independent of NPC1 or not. Thus, we demonstrate a heretofore unanticipated step in intracellular cholesterol trafficking, namely the obligate interaction of NPC2 and LBPA, and go on to provide molecular level details of the NPC2-LBPA interaction that have implications for potential therapeutic application of the findings (e.g. that PG treatment/LBPA enrichment can potentially be used in NPC2 cases where mutations are outside the identified LBPA interaction domain). We hope we have now done a better job of highlighting the new step in intracellular sterol trafficking that we’ve found, secondarily supporting the hypothesis that LBPA-stimulated sterol egress can be NPC1-independent, but only if there is functional NPC2 available.

We do not have in hand the NPC1 construct with which to repeat the experiments of Wang with our NPC2 mutants, however this is certainly planned for ongoing experiments. Please note that the assay used in Wang is essentially an equilibrium distribution analysis, with time courses shown in minutes, much faster than the rates of NPC2-mediated sterol transfer (seconds), particularly in the presence of membrane LBPA.

3) Does binding to LBPA affect the binding to cholesterol? According to the author's model, one might expect that LBPA might displace the bound cholesterol from NPC2 so that it can be transferred to liposomes.

This is an interesting question. We don’t think that LBPA is competitively displacing cholesterol from the NPC2 binding site, as follows: The tertiary structure of holo-NPC2 with bound cholesteryl sulfate (Friedland et al., 2003) shows a hand-in-glove fit for the sterol which would not likely accommodate a diacylated phospholipid. Moreover, a substrate specificity analysis using two different binding assays from the Lobel lab (Liou et al., 2006) shows not a single phospholipid being bound to NPC2, including LBPA. Rather than a displacement, we hypothesize that the interaction of the NPC2 hydrophobic knob with LBPA (a) positions the protein with its ligand binding pocket right at the membrane surface, and (b) may cause a conformational change in the binding pocket such that the cholesterol off-rate is markedly increased, as we found in our kinetics analyses. This is now more clearly stated in the Discussion section of the revised manuscript.

Reviewer #3:

This is a well-documented paper on the interaction of NPC2 protein with LBPA (otherwise known as BMP). However, given the previous publications from this group and others, I don't find the results particularly novel. It is more like working out some additional details from further mutations. Most importantly, I do not get from this paper a clear understanding of how NPC2 binding to LBPA actually leads to enhanced efflux of cholesterol from LE/LY with or without NPC1.

We thank the reviewer for considering our report on the heretofore unknown interaction between NPC2 and LBPA in intracellular cholesterol trafficking to be well documented. We respectfully disagree, however, that the manuscript is simply a repetition with some extension of our earlier mutagenesis work; the work published in McCauliff et al., (2015) focused on the surface domains of NPC2 that were necessary for cholesterol transport to zwitterionic EPC vesicles and did not include LBPA! It wasn’t until subsequent investigations that we realized that only one of the domains identified in that paper was involved in the dramatic stimulation effect of LBPA on cholesterol transfer rates by NPC2 (Xu et al., 2008). Those subsequent experiments, using LBPA-containing vesicles (and reported herein) initiated our efforts to understand the molecular basis of the effect of LBPA on cholesterol clearance in NPC1-deficient cells, which had been shown a decade ago (Chevallier et al., 2008); these studies ultimately led to the novel discovery that not only does NPC2 directly interact with LBPA in membranes, but that the mechanism of LBPA action is critically dependent on functional NPC2 protein. In order to emphasize these points, we have trimmed and modified the Introduction and Discussion section and altered the Abstract and title. We hope we have now done a better job of explaining what we consider to be a novel aspect of intracellular cholesterol trafficking within the LE/LY compartment.

Essential revisions:1) The mouse NPC2 sequence in Figure 1 shows a glutamate in the middle of the hydrophobic knob sequence. Is the mouse protein defective in transport? If not, why?

The mouse NPC2 protein does indeed have a glutamate residue (E60) within the hydrophobic domain. We did not focus on this residue in our mutagenesis studies as we chose to primarily focus on highly conserved residues within the hydrophobic knob region and, as Figure 1 shows, this is not one of the most highly conserved, nor does it have an appreciable hydrophobicity score. Additionally, we have previously shown that both human and bovine NPC2 proteins have normal cholesterol transport properties (Cheruku et al., 2006; Xu et al., 2008), each with a methionine in this position (Figure 1), and that incorporation of LBPA into membranes elicits the same change in cholesterol transfer rate by human NPC2 (Xu et al., 2008) that we show for the mouse form of the protein, suggesting that this particular residue is not integral to the function of the domain. However, what our present work emphasizes is not the individual residues within the knob that are necessary for cholesterol transport and LBPA binding by NPC2, but the region itself. We have modified our discussion to make the point that these studies suggest that this hydrophobic knob domain, which also overlaps with the NPC1 interactive domain, is highly critical for the function of NPC2.

2) In measuring the amounts of filipin labeling (Figure 6), they should measure the filipin intensity rather than the area of filipin labeling. This would be a much better indicator of the amount of cholesterol in the membranes.

As addressed to reviewer #1 above, we thank you for pointing out this error in our text. The data presented do indeed represent the intensity of the filipin stain, and appropriate changes have been made.

3) The same is true for measuring LBPA (Figure 7). Intensity measurements are made easily with any image analysis software package.

The Materials and methods section and figure legend text for all studies measuring immunofluorescence in cells have been updated to reflect that the data are representative of the fluorescence intensity per unit area.

4) Much of the Discussion section seemed repetitive of earlier sections. More importantly, it did not provide mechanisms that I understood for enhanced cholesterol efflux when LBPA and/or NPC2 levels were increased.

We have removed and/or altered some of the wording in the Results section that were repetitive in the Discussion section.

5) How would membrane:membrane interactions of LBPA-enriched membranes facilitate efflux? Most papers have suggested that LBPA is mainly on internal membranes and not on the limiting membrane. Contacts with the limiting membrane would need to clear areas of the glycocalyx, which is considered to protect the limiting membrane against hydrolysis. Merely being near the limiting membrane but bound to a LAMP protein would still leave a large aqueous gap to be traversed by cholesterol.

This is clearly the next key question — how does the increased LBPA content lead to cholesterol clearance from the LE/LY compartment? What we have learned here is that in order for the LBPA to function, it must interact with NPC2. In the revised manuscript, we include additional indirect supporting evidence from the literature for this association, regarding NPC2 localization in cells from mice with NPC1 disease, and further discuss potential mechanisms, including our recent findings of increased macro autophagy in LBPA-enriched NPC1 deficient cells. Finally, we have proposed that, while a secondary mechanism of cholesterol efflux may function independently of NPC1, the contribution of this pathway may be either very small or possibly non-existent under “normal” conditions, but in NPC1 disease the accumulation of LBPA may be a compensatory mechanism aimed at increasing the NPC2mediated transfer of cholesterol out of the inner LE/LY membranes, thereby revealing the presence of this secondary pathway of LE/LY cholesterol egress.

We hope that these changes have more clearly defined how the results support our conclusion that there exists a highly specialized domain on the NPC2 protein which specifically interacts with LBPA, and that this obligate interaction is required for normal LE/LY cholesterol export. Further, the exploitation of this interaction, via increases in LBPA levels, may be used not only in NPC1-deficient cells but also in functional mutations in NPC2 where the mutation is outside of the hydrophobic knob.

[Editors' note: the author responses to the re-review follow.]

Essential revisions:1) Please show quantitation of at least some of the key NPC2 mutants (vs. wild type) in terms of their actual endocytosed levels and correlated abilities to correct the filipin staining phenotype shown in Figure 9.

The reviewers request quantification of the amount of WT or mutant NPC2 protein taken up by cells and correlation with cholesterol accumulation (Figure 9). We’d like to respectfully point out, however, that unlike experiments where plasmids encoding for various WT and mutant proteins are transfected and where non-equivalent expression levels can be an important variable, we used purified NPC2 proteins (not conditioned media) and added the identical amounts of each protein to identical cultures of NPC2 deficient cells. Further, we draw attention to prior literature in which investigators incubated NPC2 proteins with NPC2-deficient cells. In none of the following reports, which to our knowledge are the only studies published, was quantification of the amount of different proteins taken up performed:

1) Ko et al., (2003), WT NPC2 and point mutants

2) Liou et al., (2006), different NPC2 glycoforms

3) Wang et al., (2010) WT and NPC2 point mutants

4) McCauliff et al., (2015) WT and NPC2 point mutants

Our reasonable assumption is that the different proteins are taken up equivalently, given (1) the proteins all fold properly as evidenced by their cholesterol binding affinities, thus their shapes and sizes are virtually if not entirely identical and (2) the uptake of exogenous proteins, and even other macromolecules added to media such as cyclodextrins, into fibroblasts occurs via endocytosis. Since the fibroblasts themselves are not variable, all the cell culture wells are reasonably assumed to endocytose equal amounts of culture media. We have now included these assumptions in the manuscript.

2) Please discuss more fully the NPC1-dependence or -independence of the LBPA/NPC2 pathway. A lingering concern is the relevance of this NPC2/LBPA pathway to cholesterol trafficking. In response to reviewer 3's point 5, the authors say that "the contribution of this pathway may be either very small or possibly non-existent under "normal conditions, but in NPC1 disease the accumulation of LBPA may be a compensatory mechanism...". Yet, the Title says "Intracellular cholesterol trafficking is dependent upon NPC2 interaction with LBPA". Please discuss more clearly in the Discussion section. The authors say that studying the effects of the LBPA-sensitive NPC2 mutants on NPC1-mediated cholesterol transfer is tangential to their main point that LBPA acts through NPC2. However, their conclusions would be strengthened by studying these mutants to address NPC1-independence or dependence (OPTIONAL). "Potential NPC2 binding site on NPC1"? Li et al., 2016 should also be cited in the Introduction; Deffieu et al., 2011 showed direct binding of NPC2 to NPC1 and Li et al., 2016 measured Kds for this interaction and defined corresponding residues needed for the interaction. The binding site is pretty clear from the crystal structures and mutagenesis, or?

Our studies show that egress of cholesterol from the endolysosomal compartment does indeed appear to be dependent upon the interaction between NPC2 and LBPA. While normal LE/LY cholesterol efflux is also dependent upon functional NPC1 protein, our studies have additionally indicated, as have others (Boadu et al., 2012; Goldman and Krise, 2010; Kennedy et al., 2012), that cholesterol can pass the limiting lysosomal membrane in the absence of functional NPC1 in a yet to be fully determined pathway; it is the contribution of this NPC1-independent pathway that we are referring to. In our previous letter, we indeed indicated that we are unable to determine whether or not this NPC1-independent pathway functions under normal conditions or is a compensatory mechanism for correcting cholesterol accumulation in NPC1 disease. In a non-disease state, where both NPC1 and NPC2 proteins are functional, it seems likely that the main pathway of cholesterol egress involves (1) an obligate interaction between NPC2 and LBPA to transfer cholesterol from inner-LE/LY membranes to NPC2 and (2) “hydrophobic handoff” of cholesterol from NPC2 to NPC1 at the limiting membrane with NPC1 facilitating final stage of efflux from the endolysosomal compartment. We apologize if the statements made in our letter caused confusion, and have modified the discussion to ensure that the dependence of intracellular cholesterol trafficking on the NPC2-LBPA interaction is clear. Again, our main point is not the existence of a potentially NPC1-independent mechanism of cholesterol egress; the highly novel aspect of our findings is that there is an obligate interaction between NPC2 and LBPA that is required for cholesterol egress from the LE/LY.

The additional citation (Li et al., 2016) has been added in reference to NPC2-NPC1 interaction. Thank you for pointing this out.